# Human cytomegalovirus long noncoding RNA4.9 regulates viral DNA replication

Julie Tai-Schmiedel[1], Sharon Karniely[2]*, Betty Lau[3], Adi Ezra[1], Erez Eliyahu[1], Aharon Nachshon[1], Karen Kerr[3], Nicolás Suárez[3], Michal Schwartz[1], Andrew J. Davison[3], Noam Stern-Ginossar[1]*

**1** Weizmann Institute of Science, Department of Molecular Genetics, Rehovot, Israel, **2** Kimron Veterinary Institute, Bet Dagan, Israel, **3** MRC-University of Glasgow Centre for Virus Research, Glasgow, United Kingdom

☉ These authors contributed equally to this work.
* sharonkarniely@gmail.com (SK); noam.stern-ginossar@weizmann.ac.il (NSG)

**Data Availability Statement:** All relevant data are within the manuscript and its Supporting Information files. The sequencing data associated with this study are available in GSE134423.

## Abstract

Viruses are known for their extremely compact genomes composed almost entirely of protein-coding genes. Nonetheless, four long noncoding RNAs (lncRNAs) are encoded by human cytomegalovirus (HCMV). Although these RNAs accumulate to high levels during lytic infection, their functions remain largely unknown. Here, we show that HCMV-encoded lncRNA4.9 localizes to the viral nuclear replication compartment, and that its depletion restricts viral DNA replication and viral growth. RNA4.9 is transcribed from the HCMV origin of replication (*oriLyt)* and forms an RNA-DNA hybrid (R-loop) through its G+C-rich 5' end, which may be important for the initiation of viral DNA replication. Furthermore, targeting the RNA4.9 promoter with CRISPR-Cas9 or genetic relocalization of *oriLyt* leads to reduced levels of the viral single-stranded DNA-binding protein (ssDBP), suggesting that the levels of ssDBP are coupled to the *oriLyt* activity. We further identified a similar, *oriLyt*-embedded, G+C-rich lncRNA in murine cytomegalovirus (MCMV). These results indicate that HCMV RNA4.9 plays an important role in regulating viral DNA replication, that the levels of ssDBP are coupled to the *oriLyt* activity, and that these regulatory features may be conserved among betaherpesviruses.

## Author summary

Viruses have efficiently organized genomes, mostly consisting of coding genes. Nonetheless, in recent years it became apparent that herpesviruses encode for long non-coding RNAs (lncRNAs). In this study, we show that one of human cytomegalovirus (HCMV) encoded lncRNAs, named RNA4.9, is important for viral DNA replication and viral propagation. RNA4.9 is embedded in the viral origin of replication and its transcription causes the formation of a RNA-DNA hybrid, a structure which is likely important for the viral origin of replication unwinding and initiation of viral DNA replication. Furthermore, interfering with viral origin of replication or with RNA4.9 promoter activities leads to reduced levels of the viral single-stranded DNA-binding protein (ssDBP), suggesting that

**Funding:** This research was supported by ICORE grant (Chromatin and RNA Gene Regulation awarded to N.S-G.), the Israeli Science Foundation (1526/18 awarded to N.S-G.) and the Medical Research Council (MC_UU_12014/3 awarded to A. J.D). N.S-G is an incumbent of the Skirball Career Development Chair in New Scientists. The funders had no role in study design, data collection and analysis, decision to publish, or preparation of the manuscript.

**Competing interests:** The authors have declared that no competing interests exist.

the ssDBP levels are coupled to the origin activity. Finally, we discovered a new lncRNA encoded by the murine cytomegalovirus, which seems to have similar features and function as the HCMV encoded RNA4.9. These results suggest a novel mechanism, conserved among betaherpesviruses, by which a viral lncRNA, embedded in the viral origin of replication, regulates viral DNA replication and may play a role in coupling origin activity with the level of ssDBP.

## Introduction

The emergence of genome wide high-throughput sequencing technology revealed the intriguing complexity of the human transcriptome and the existence of thousands of long non-coding RNAs (lncRNAs), which are processed similarly to mRNAs but appear not to give rise to functional proteins [1]. Although an increasing number of lncRNAs are implicated in a variety of cellular functions, they do not form a well-defined class of transcripts that act through a common pathway. Thus, most lncRNAs remain poorly characterized mechanistically. The few well-studied examples include lncRNAs that act in the nucleus and regulate gene expression in *cis* or in *trans* through recruitment of proteins or molecular complexes to specific loci [1,2]. LncRNAs can also act as scaffolds that bring together different proteins or bridge protein complexes and specific chromatin regions [3]. In addition, there is a growing list of assigned functions for mature cytoplasmic lncRNAs, such as regulation of translation by hybridization to target mRNAs, functional modulation of cytosolic proteins, and acting as decoys for short RNAs or RNA-binding proteins [4,5].

Members of the family *Herpesviridae* are large DNA viruses that infect a wide range of vertebrates, including humans. They typically cause acute disease associated with lytic infection, followed by benign, life-long persistence involving latent infection with occasional reactivation [6]. Although viruses are known for their compact genomes in which regions not encoding proteins are rare, a number of highly expressed lncRNAs have been identified in herpesviruses and shown to have critical roles. These roles include: regulation of chromatin structure [7], latency establishment, maintenance and reactivation [8–10], recruitment of cellular transcription factors to viral DNA [11], inhibition of virus-induced apoptosis [12,13], and quenching of cellular miRNAs [14,15]. Human cytomegalovirus (HCMV) is a ubiquitous member of the subfamily *Betaherpesvirinae*, infecting the majority of human population worldwide and leading to severe diseases in newborns and immunocompromised adults. The viral genome is 236 kbp in size and has been estimated to contain at least 170 open reading frames encoding functional proteins [16,17]. Recent studies revealed a more complex expression pattern, characterized by many previously undetected transcribed and translated sequences during lytic and latent infection [18–22]. In particular, RNA sequencing (RNA-seq) experiments have shown that the majority of polyadenylated viral RNA transcription is committed to the production of four lncRNAs (RNA2.7, RNA1.2, RNA4.9, and RNA5.0; the numbers represent transcript lengths in kb) [19]. These lncRNAs are expressed by low- and high-passage HCMV isolates during lytic and latent infection [20,21,23]. Although their high expression levels and widespread occurrence suggests an important function in viral propagation, the functions of these viral lncRNAs remain largely unknown.

Here, we have analyzed the expression and localization of the four major HCMV lncRNAs. Whereas RNA1.2, RNA2.7 and RNA5.0 are distributed in the cytoplasm, RNA4.9 is localized to the nuclear viral replication compartment. We found that ablating RNA4.9 expression inhibited viral DNA synthesis and consequently reduced viral titers. We further demonstrate

that RNA4.9 forms an R-loop involving the unusually G+C-rich region at its 5' end and that it regulates viral DNA replication and also effects the levels of the HCMV single-stranded DNA-binding protein (ssDBP; encoded by gene UL57). Moreover, we discovered that murine cytomegalovirus (MCMV) encodes a lncRNA with a similar genomic localization and G+C-composition to those of HCMV RNA4.9. These results suggest a novel mechanism, conserved among betaherpesviruses, by which a viral lncRNA, embedded in the origin of viral DNA replication (*oriLyt*), regulates viral DNA replication and may play a role in coupling *oriLyt* activity with the level of ssDBP.

## Results

### RNA4.9 is concentrated in the viral replication compartment

Our measurements of viral RNA levels during HCMV infection [24] revealed that the levels of HCMV lncRNAs are very high throughout lytic infection of fibroblasts. At 72 hours post infection (hpi), their expression is higher than most viral genes and they greatly outnumber (up to 190- and 95-fold for RNA2.7 and RNA1.2, respectively) the levels of the abundant cellular transcript *actinB* (Fig 1A). Since cellular lncRNAs have diverse molecular functions in various cellular compartments, we first investigated the subcellular localization of the HCMV lncRNAs. Using fluorescence *in situ* hybridization (FISH) in HCMV-infected fibroblasts, we found that RNA1.2, RNA2.7 and RNA5.0 localize almost exclusively to the cytoplasm. In contrast, RNA4.9 concentrates in discrete sub-nuclear sites reminiscent of viral DNA replication compartments (Fig 1B). The detected signal was confirmed as viral and RNA-specific, since it was detected only in infected cells (S1A Fig) and it was eliminated by pre-treatment of cells with RNase A and not DNase I (S1B Fig). Combining FISH for RNA4.9 with immunofluorescence (IF) staining of the viral DNA polymerase processivity subunit (encoded by gene UL44), we confirmed that RNA4.9 resides within viral DNA replication compartments (Fig 1C). In addition, metabolic labeling of nascent viral DNA labeled with ethynyl-2′-deoxyuridine (EdU) and visualized using "Click" chemistry [25] further illustrates that RNA4.9 localizes in the replication compartment (Fig 1D). To establish further the nuclear localization of RNA4.9, we performed subcellular fractionation and analyzed RNA 4.9 expression in the nuclear and cytoplasmic fractions. In agreement with FISH staining, RNA4.9 was enriched in the nuclear fraction, whereas RNA2.7 was mostly cytosolic (S1C Fig).

### Interference with RNA4.9 expression inhibits viral DNA replication and viral titers

The localization of RNA4.9 to the viral replication compartment hinted to a possible role in DNA replication and motivated us to assess its role during infection. Analysis of RNA4.9 expression kinetics showed that, in agreement with our RNA-seq measurements, RNA4.9 was expressed already at 5 hpi and continually raised as infection progressed (Fig 2A). Inhibition of viral DNA replication using viral DNA polymerase inhibitor phosphonoformic acid (PFA) reduced RNA4.9 expression, most likely due to the decrease in available viral genomic template, but did not abolish it (Fig 2A). We next asked if the sub-nuclear localization of RNA4.9 is dependent on viral DNA synthesis by blocking viral DNA synthesis and analyzing RNA4.9 and UL44 expression at 48 hpi using FISH and IF, respectively. RNA4.9 levels were reduced by inhibition of viral DNA replication but concentration of RNA4.9 in sub-nuclear sites was not affected by PFA treatment (Fig 2B). However, in agreement with previous observations [26], inhibition of viral DNA replication resulted in dispersion of UL44 in the nucleus (Fig 2B).

a

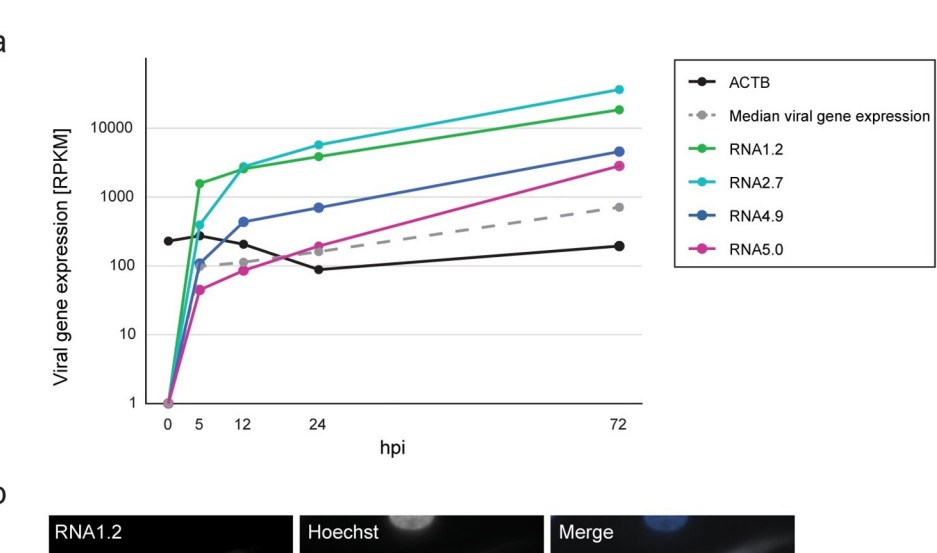

b

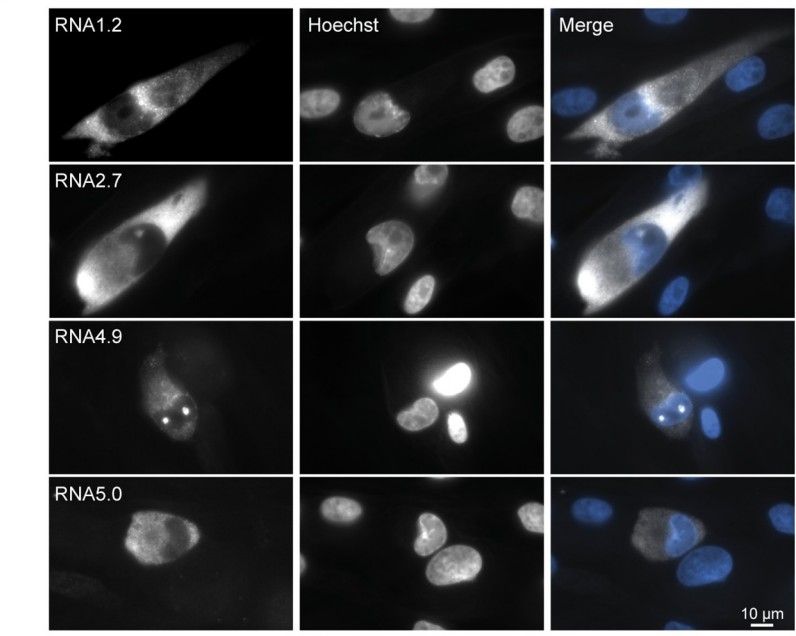

c

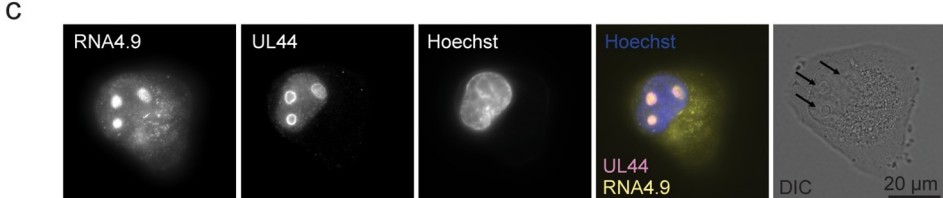

d

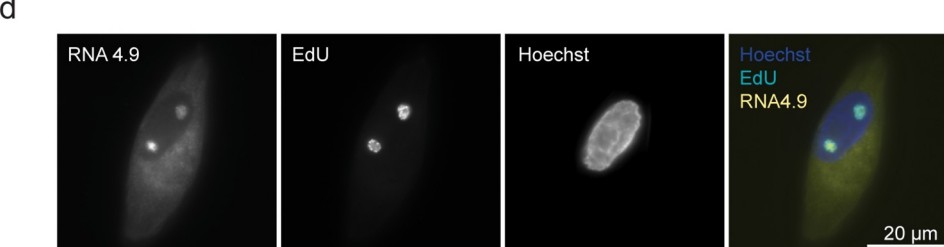

**Fig 1. Expression kinetics and subcellular localization of HCMV lncRNAs. a)** Expression levels of HCMV encoded lncRNAs (RNA2.7, RNA1.2, RNA4.9 and exonic RNA5.0) together with the median expression of viral genes and one host transcript (ACTB) as measured by RNA-seq during HCMV infection in fibroblasts (MOI = 5) [24]. **b)** HCMV lncRNAs were detected by RNA-FISH using fluorescent probes (white) in HCMV Merlin strain-infected fibroblasts at 48 hpi (MOI = 1). **c)** RNA4.9 and the UL44 protein were detected in HCMV Merlin strain-infected fibroblasts at 48 hpi (MOI = 5) using RNA-FISH and IF, respectively. Differential interference contrast (DIC) of the stained cell shows viral DNA replication compartments, indicated by black arrows. **d)** RNA4.9 and nascent DNA were detected in HCMV Merlin strain-infected fibroblasts at 48 hpi (MOI = 3) using RNA-FISH and EdU incorporation followed by labelling with a 6-FAM fluorescent azide using the "Click" chemistry. **b-d)** Nuclei were counterstained with Hoechst (blue, in merge).

Taken together, these results indicate that the initial expression and localization of RNA4.9 occurs independent of viral DNA synthesis.

Complete deletion of the RNA4.9 gene leads to a non-infectious virus, since it overlaps essential regions of *oriLyt* [27,28]. Therefore, to assess RNA4.9 function, we attempted to knock down (KD) RNA4.9 expression using siRNAs. We transfected fibroblasts with siRNAs targeting RNA4.9 or RNA1.2 (as a control) and infected them with virus for 48 hours. RNA1.2 expression was significantly reduced, but there was no significant reduction in RNA4.9 expression (S2A Fig), probably because the RNAi machinery is located mainly in the cytoplasm and access to nuclear-retained lncRNAs is limited [29]. We further attempted to use chemically modified chimeric DNA antisense oligonucleotides (ASO), which have been used to deplete nuclear RNAs, but, in accordance with reports that ASOs are often inefficient [30], we were not able to ablate RNA4.9 expression (S2B Fig).

We then applied two different approaches aimed at inhibiting RNA4.9 expression while minimally interfering with the integrity of *oriLyt*. The first used the CRISPR–Cas9 system to create deletion mutations around the RNA4.9 promoter. The second used CRISPR interference (CRISPRi), in which a nuclease-inactive derivative of Cas9 is fused to the KRAB repressor domain (dCas9), to bind near the RNA4.9 promoter and block RNA4.9 transcription without introducing mutations [31]. In each approach, two sgRNAs mapping close to the RNA4.9 transcriptional start site (TSS) were selected, and their use resulted in significant RNA4.9 KD (Fig 2C and S2C Fig). In both approaches, a control sgRNA was used that did not cause RNA4.9 KD. RNA4.9 KD using both approaches led to reduced viral DNA synthesis (Fig 2D and S2D Fig) and viral growth (Fig 2E and S2E Fig), but the KD efficiency and the corresponding effects were much greater using CRISPR–Cas9. No effect on expression of immediate early (IE) gene UL123 (encoding IE1) was observed, and a minimal reduction in early gene UL44 expression was detected at 24 hpi, when DNA replication starts. However, expression of early gene UL44 and late gene UL99 was reduced substantially at 48 hpi, when extensive DNA replication takes place (Fig 2F and S2F Fig). These results suggest that RNA4.9 expression is important for viral DNA synthesis and growth.

## Targeting the RNA4.9 promoter with CRISPR-Cas9 directly inhibits viral DNA replication

HCMV DNA replication is dependent on early gene expression because it relies on early gene products. In addition, DNA replication affects the levels of early gene expression, as increased levels of viral DNA give rise to more viral transcripts. Therefore, we investigated whether interference with RNA4.9 expression affects early viral gene expression directly, leading to a secondary reduction in viral DNA replication, or whether it affects DNA replication, leading to a secondary effect on early gene expression. This was done by infecting control cells and cells in which RNA4.9 was targeted by CRISPR (RNA4.9-Cas9 KD), in the presence of PFA, which blocks viral DNA synthesis, thus ensuring that any effect on gene expression was directly due

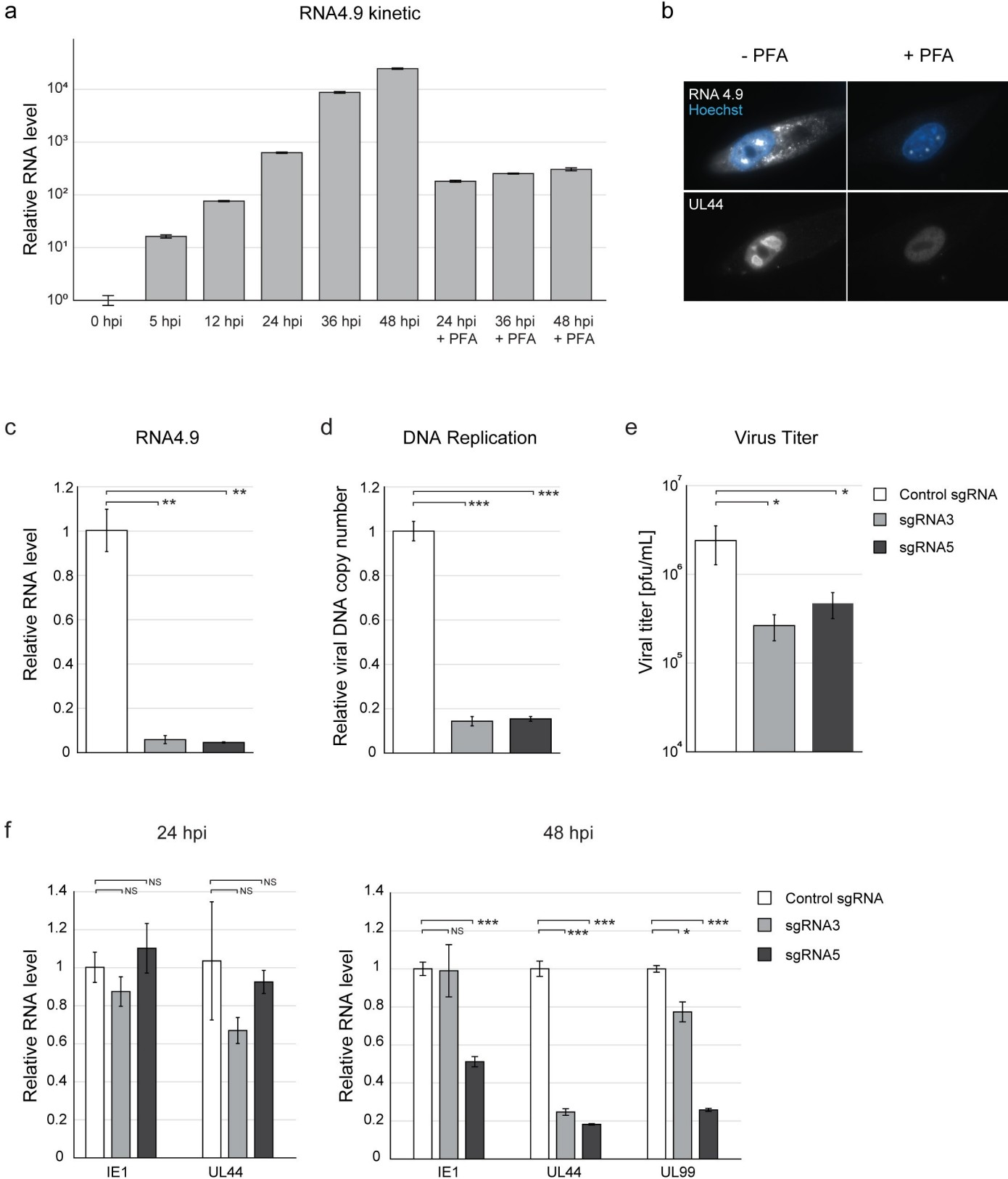

**Fig 2. RNA4.9 shows immediate early kinetics and its KD inhibits HCMV DNA replication and growth. a)** HCMV Merlin strain-infected fibroblasts (MOI = 1) were harvested at indicated time points. Infected cells were treated with the viral DNA replication inhibitor phosphonoformic acid (PFA) as indicated. Relative levels of RNA4.9 were quantified using RT-qPCR and normalized to the cellular transcript ANXA5. **b)** RNA4.9 and the UL44 protein (white) were detected in HCMV Merlin strain-infected fibroblasts at 48 hpi (MOI = 5) using RNA-FISH and IF, respectively. Infected cells were treated with PFA as indicated. Nuclei were counterstained with Hoechst (blue). **c-f)** Fibroblasts expressing CAS9 and either a control sgRNA or one of two different sgRNAs targeting the RNA4.9 promoter (sgRNA3 and sgRNA5) were infected with HCMV Merlin strain (MOI = 0.1). **c)** Relative levels of RNA4.9 were quantified using RT-qPCR at 48 hpi and normalized to the cellular transcript ANXA5. **d)** Relative viral DNA levels were quantified using qPCR at 48 hpi using UL44 primers, and normalized to the cellular gene B2M. **e)** Viral titers were measured 5 days post infection (dpi) by TCID50. **f)** Relative levels of HCMV immediate early gene UL123 (encoding IE1) and early gene UL44 were quantified by RT-qPCR at 24 and 48 hpi, and late gene UL99 was quantified at 48 hpi. RNA levels were normalized to the cellular ANXA5 transcript. **c-f)** Values and error bars represent the mean and SD of triplicates. A representative analysis of two independent experiments is shown. Two-sided $t$-test was applied (***p-value<0.001, **p-value<0.01, *p-value<0.05, NS, not significant).

to RNA4.9 depletion. Real-time PCR and RNA-seq analysis demonstrated that RNA4.9-Cas9 KD did not result in any other major changes (greater than 2-fold) in viral gene expression (Fig 3A, S3 Fig and S1 Table), when viral DNA replication was inhibited. These results suggest that RNA4.9 is involved in viral DNA replication, and that the effect on viral gene expression is a result of the availability of fewer viral template DNA molecules.

## The G+C-rich region of RNA4.9 forms RNA-DNA hybrids within *oriLyt*

The first 800 nucleotides of RNA4.9 are highly G+C-rich (~80%). G+C-rich islands are often found adjacent to origins of DNA replication in nuclear and mitochondrial genomes, as well as in viral DNA [32–34]. Transcripts originating from these sites are known to form triplex RNA-DNA structures (R-loops) with their DNA template and to be functionally important for the initiation of DNA replication [35,36]. In cells infected by a gammaherpesvirus, Epstein-Barr virus (EBV), viral replication depends on the formation of an RNA-DNA hybrid at *oriLyt* by the viral G+C-rich BHLF1 transcript [37]. Similarly, the G+C-region of an *oriLyt*-associated transcript is essential for DNA replication of another gammaherpesvirus, Kaposi's sarcoma-associated herpesvirus (KSHV) [38]. RNA-DNA hybrids were previously described within HCMV OriLyt [39]. We therefore examined whether RNA4.9 forms an RNA-DNA hybrid through its G+C-rich domain, using DNA-RNA hybrid immunoprecipitation (DRIP). DNA fragments containing the G+C-rich domain of the RNA4.9 gene were enriched in the pulled down fraction similar to the cellular gene APOE, which is known to form R-loops [40,41]. In contrast, DNA fragments encompassing the RNA1.2 and RNA2.7 genes were not enriched in the pulled down fraction similar to nonrelated genomic regions (Fig 3B). These results suggest that RNA4.9 forms an RNA-DNA hybrid via its G+C-rich domain.

## Interfering with *oriLyt* activity reduces HCMV ssDBP protein levels

In EBV, formation of an RNA-DNA hybrid at *oriLyt* is required for recruitment of ssDBP to *oriLyt* [37]. This prompted us to test whether RNA4.9-Cas9 KD affects the recruitment of HCMV ssDBP, encoded by the UL57 gene, to viral DNA replication sites. At 24 and 48 hpi, immunofluorescence analysis revealed that ssDBP was concentrated at specific foci representing viral DNA replication sites, as reported previously ([42]; Fig 4A and 4B). However, RNA4.9 KD resulted in a considerable reduction in the ssDBP signal (Fig 4A and 4B). A similar reduction in ssDBP signal was observed when these experiments were conducted in the presence of PFA, thus precluding an indirect effect caused by reduction of viral DNA replication (Fig 4C). To test the specificity of these effects, we analyzed the expression and localization of the UL84 protein in these cells, as it has been proposed that this protein binds *oriLyt* via an RNA stem-loop [43,44]. However, RNA4.9-Cas9 KD in the presence of PFA did not lead to major changes in the amount or localization of the UL84 protein (S4A Fig).

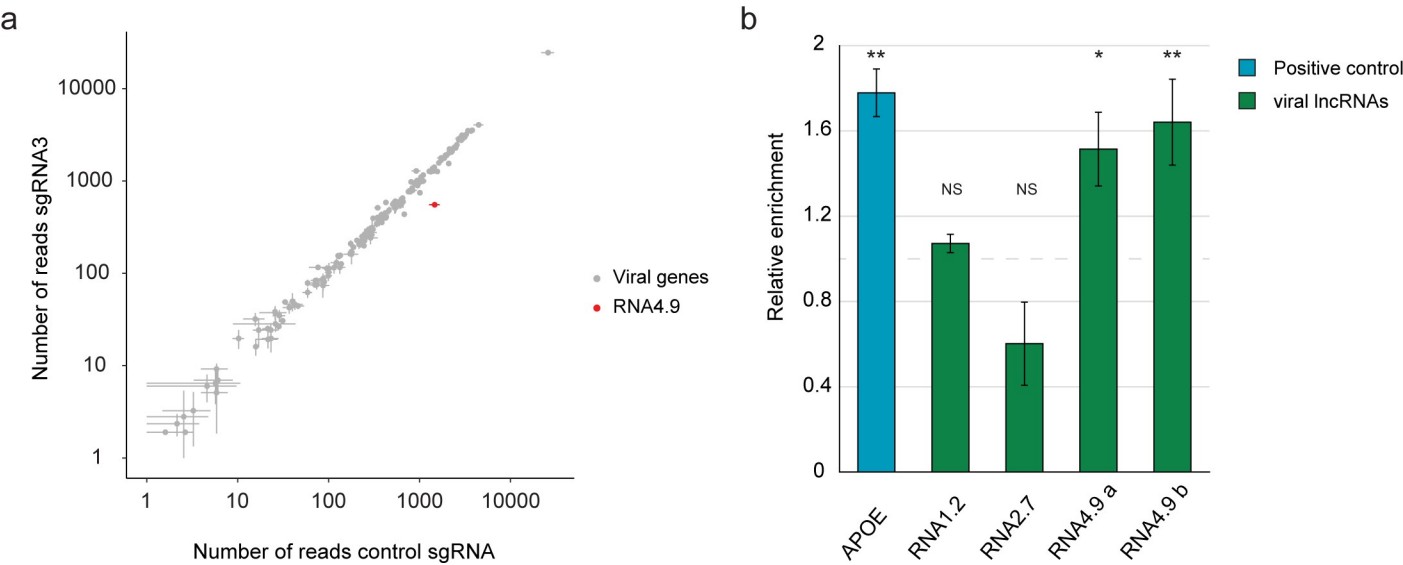

**Fig 3. RNA4.9 forms R-loops and its Cas9 KD does not directly affect viral gene expression. a)** Scatter plot presenting canonical viral gene expression in fibroblasts expressing CAS9 and a control sgRNA or a sgRNA targeting the RNA4.9 promoter (sgRNA3), which were infected with HCMV Merlin strain (MOI = 0.01), as measured by RNA-seq at 24 hpi with PFA treatment. The average and SD of two independent experiments is shown. **b)** Fibroblasts were infected with HCMV Merlin strain (MOI = 3) and cells were harvested at 48 hpi for DNA-RNA hybrid IP (DRIP). Enrichment of viral lncRNAs regions (green; RNA4.9 region was quantified using two sets of primers a and b) and an R-loop-forming cellular gene (APOE, blue; positive control) relative to the immunoprecipitation with an isotype control was quantified by qPCR. Fold-enrichment was calculated relative to a non-R-loop-forming cellular gene (SLC22A1). A relative enrichment of 1 represents no enrichment (dashed grey line). Values and error bars represent the average and SD of triplicates. A representative analysis of two independent experiments is shown. One-sided *t*-test was applied and compared to the non-R-loop-forming cellular gene (SLC22A1), (\*\*p-value<0.01, \*p-value<0.05, NS, not significant).

The specific effect of RNA4.9-Cas9 KD on ssDBP levels was further validated by immuno-blotting. At 24 hpi, ssDBP levels were drastically reduced by RNA4.9-Cas9 KD (Fig 4D and 4E). This effect was specific to ssDBP, as no significant changes were detected in the levels of the early UL44 and UL84 proteins (Figs 4D and S4B), which participate in DNA replication. Similar levels of IE1 in RNA4.9-Cas9 KD and control cells confirmed that these cells were equally infected. At 48 hpi, RNA4.9 KD resulted in a strong reduction in the levels of the UL44 and UL84 proteins that was mostly dependent on viral DNA replication, as it did not occur in the presence of PFA (Figs 4D and S4B). In contrast, the reduction in ssDBP was mostly inde-pendent of DNA replication, as it also occurred in the presence of PFA (Fig 4D and 4E). This indicates that the effect of RNA4.9-Cas9 KD on ssDBP level is independent of the reduction in template molecules.

Since gene UL57 is located immediately next to oriLyt, on the opposite side from the RNA4.9 gene (Fig 4F), we assessed whether RNA4.9-Cas9 KD affects the levels of UL57 tran-scripts. At 24 and 48 hpi, in the presence of PFA, ssDBP levels were reduced greatly by RNA4.9-Cas9 KD (7.8- and 7.7-fold, respectively; Fig 4E). In contrast, UL57 transcript levels showed minimal reduction (1.7- and 2-fold, respectively; Fig 4G), in accordance with the RNA-seq measurements (S1 Table).

During these experiments, we noticed that our ability to detect ssDBP protein coincided with the onset of DNA replication, as we detected the ssDBP protein only at 24 hpi (Fig 5A) and significantly higher levels at 48 hpi, coinciding with the time of substantial viral DNA rep-lication. On the other hand, UL57 transcript levels did not show strong dependency on DNA replication (Figs 5B, S5A and S5B). In addition, using IF combined with FISH, we detected RNA4.9 in distinct nuclear foci at 12 hpi, whereas we were able to detect the ssDBP only at time points in which viral DNA replication likely initiated and only in proximity to RNA4.9

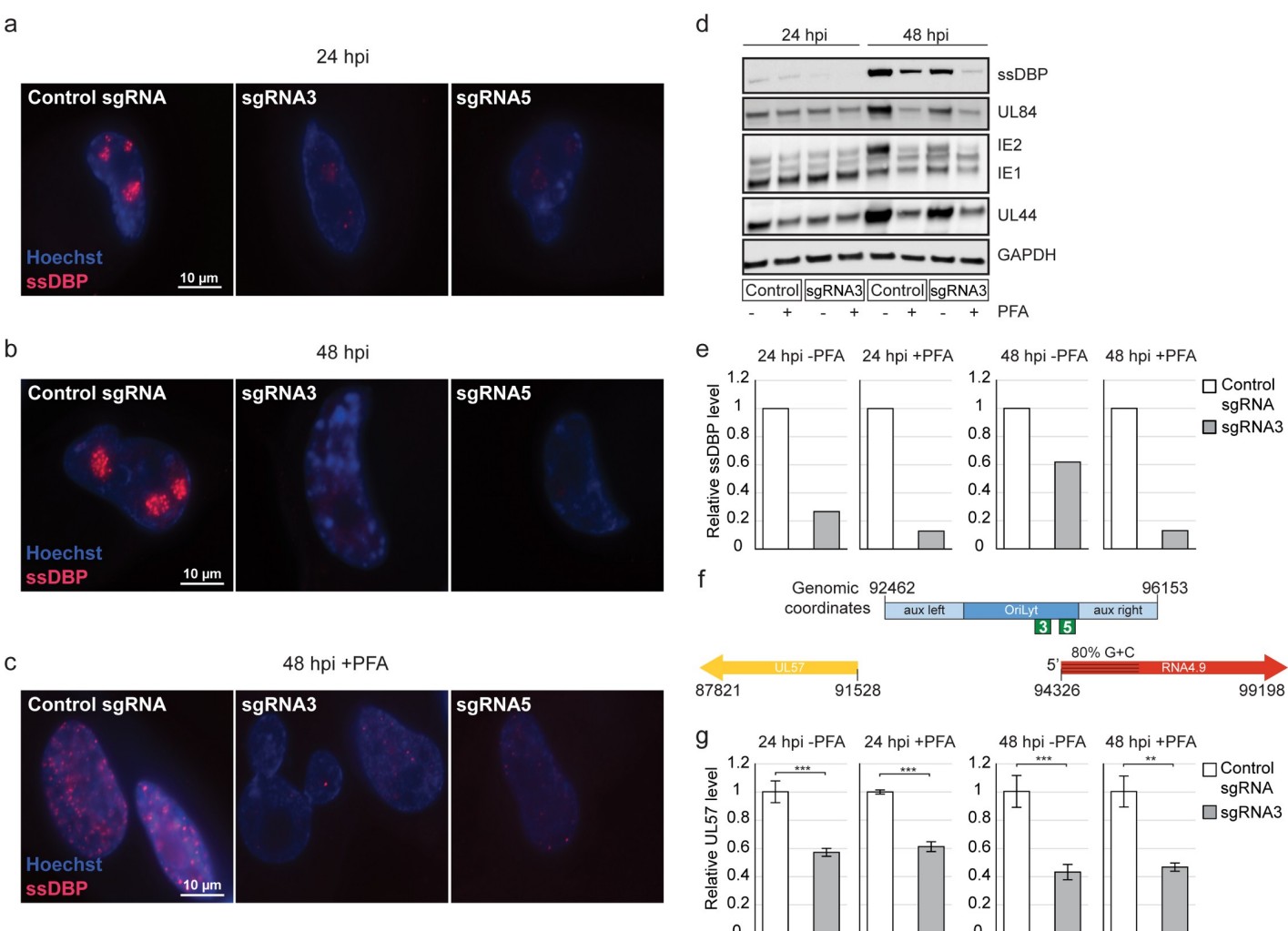

**Fig 4. RNA4.9-Cas9 KD substantially reduces ssDBP levels.** Fibroblasts expressing CAS9 and either a control sgRNA or a sgRNA targeting RNA4.9 promoter (sgRNA3 or sgRNA5, as indicated) were infected with HCMV Merlin strain (MOI = 3). **a)** ssDBP (red) was detected by IF at 24 hpi, **b)** 48 hpi or **c)** 48 hpi in the presence of PFA. Cells were counterstained with Hoechst (blue). **d)** ssDBP, IE1/IE2, UL84 and UL44 proteins were detected by immunoblot analysis at 24 or 48 hpi, with and without PFA treatment. GAPDH was used as a loading control. **e)** Quantification of ssDBP levels from the immunoblot analysis in (d), normalized to the levels of GAPDH. **f)** Schematic representation of HCMV *oriLyt* (NC_006273.2), showing the core region (blue) and flanking auxiliary regions (aux, light blue), and genes RNA4.9 (red) and UL57 (yellow). sgRNA PAM sites are indicated in green (sgRNA3–93412 and sgRNA5–93378). **g)** Relative UL57 levels were quantified using RT-qPCR at 24 and 48 hpi, with and without PFA treatment, and normalized to the cellular ANXA5 transcript. Two-sided *t*-tests were applied (***p-value<0.001, **p-value<0.01, *p-value<0.05).

(Fig 5C). Therefore, we tested the possibility that interference with RNA4.9 expression affects ssDBP stability, perhaps as an outcome of faulty unwinding of *oriLyt*. To this end, we ectopically expressed under the same promotor either ssDBP fused to a flag-tag (ssDBP-flag) or mCherry as a control, and measured protein levels during HCMV infection. mCherry expression was detected at 8 hpi, and the level increased moderately at 24hpi (Fig 5D). In contrast, ssDBP-flag was detected only at 24 hpi, when viral DNA replication started (Fig 5E and S5B Fig), and the level was increased considerably at 48 hpi (Fig 5E). Addition of PFA, which inhibits viral DNA synthesis by blocking the viral DNA polymerase, resulted in only a mild reduction of ectopically expressed ssDBP (Fig 5E). These results indicate that the initial steps of replication, which precede viral DNA synthesis, such as the *oriLyt* unwinding, are the ones that are likely coupled to ssDBP levels. They also suggest a possible relationship between

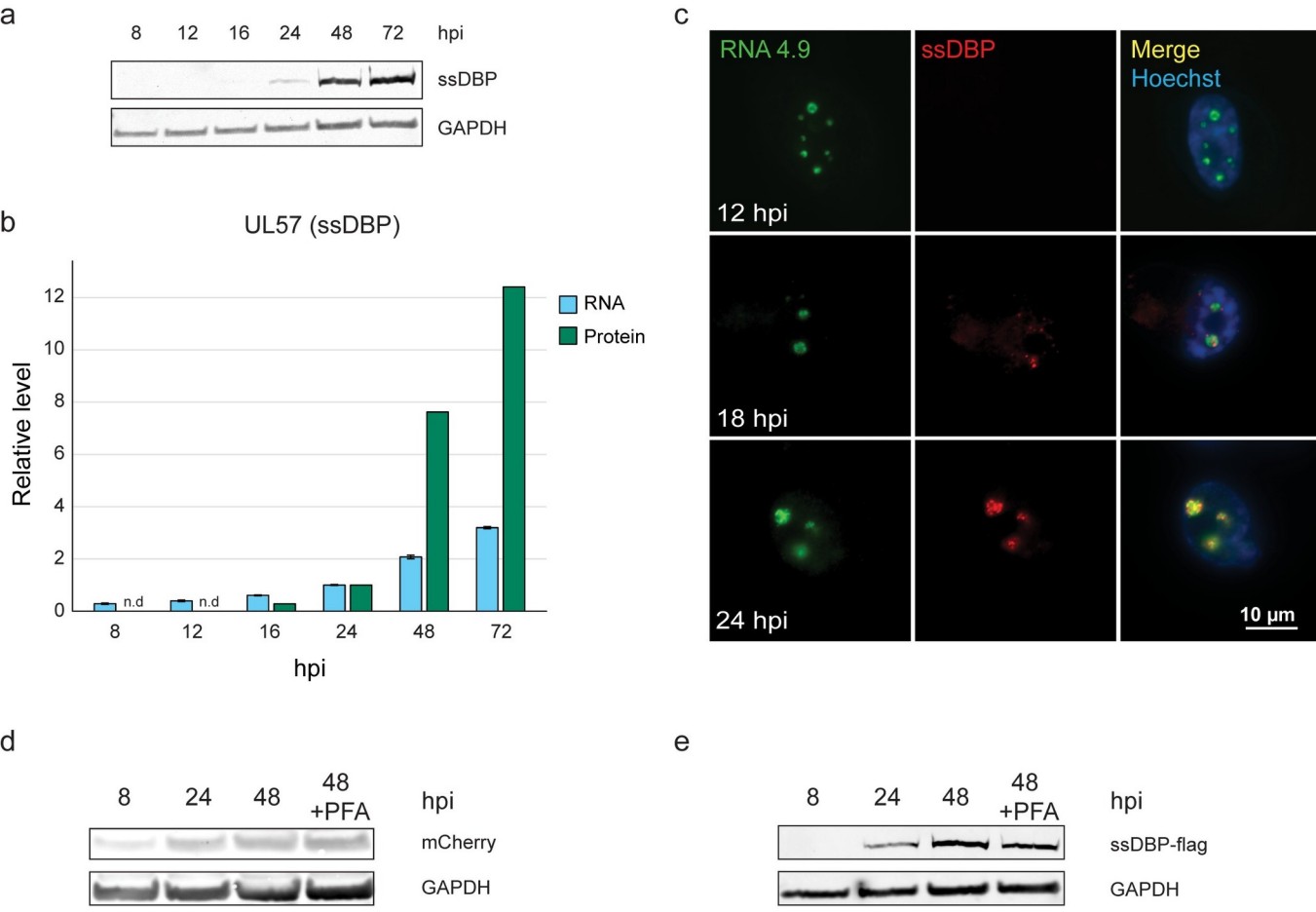

**Fig 5. ssDBP levels are linked to viral DNA replication. a-b)** Fibroblasts were infected with HCMV Merlin strain (MOI = 2) and harvested at indicated time points after infection. **a)** ssDBP was detected by western blot analysis and GAPDH was used as loading control. **b)** In green are ssDBP levels as measured by western blot presented in (a), normalized to GAPDH and in light blue are UL57 transcript levels as quantified by RT-qPCR and normalized to ANXA5. **c)** RNA4.9 (green) and ssDBP (red) were detected in HCMV Merlin strain-infected fibroblasts (MOI = 3) at indicated time points using RNA-FISH and IF, respectively. Nuclei were counterstained with Hoechst (blue). **d-e)** Fibroblasts ectopically expressing either mCherry or ssDBP-flag were infected with HCMV Merlin strain (MOI = 2), harvested at the indicated time points post infection and treated with PFA during infection as indicated. mCherry (d) and ssDBP-flag (e) were detected by immunoblot analysis using anti-mCherry and anti-Flag antibodies, respectively. GAPDH was used as loading control.

RNA4.9, *oriLyt* activity and ssDBP expression. However, since these effects were observed by targeting short regions of *oriLyt* using the Cas9 system, it is possible that the physical DNA cleavage rather than RNA4.9 transcription was responsible for inhibition of viral DNA replication and ssDBP expression. Therefore, we examined five different sgRNAs targeting the RNA4.9 TSS region (two of which, sgRNA3 and sgRNA5 were used in all the above experiments) and confirmed that all of them cleave the HCMV genome with similar efficiency (S6 Fig). Despite similar cleavage efficiencies, the three sgRNAs that triggered the most significant reduction in RNA4.9 transcription (sgRNA2, sgRNA3 and sgRNA5 in Fig 6A) were most effective in decreasing the levels of ssDBP (Fig 6B).

The results described above support a direct connection between reduction in RNA4.9 transcription, viral DNA replication and ssDBP expression. To test more directly the effect of RNA4.9 expression on DNA replication, we generated a viral mutant in which the predicted RNA4.9 TATA sequence was deleted (ΔTATA). When comparing this mutant to its parental virus in the presence of PFA, we observed a 2.5-fold reduction in RNA4.9 expression, whereas

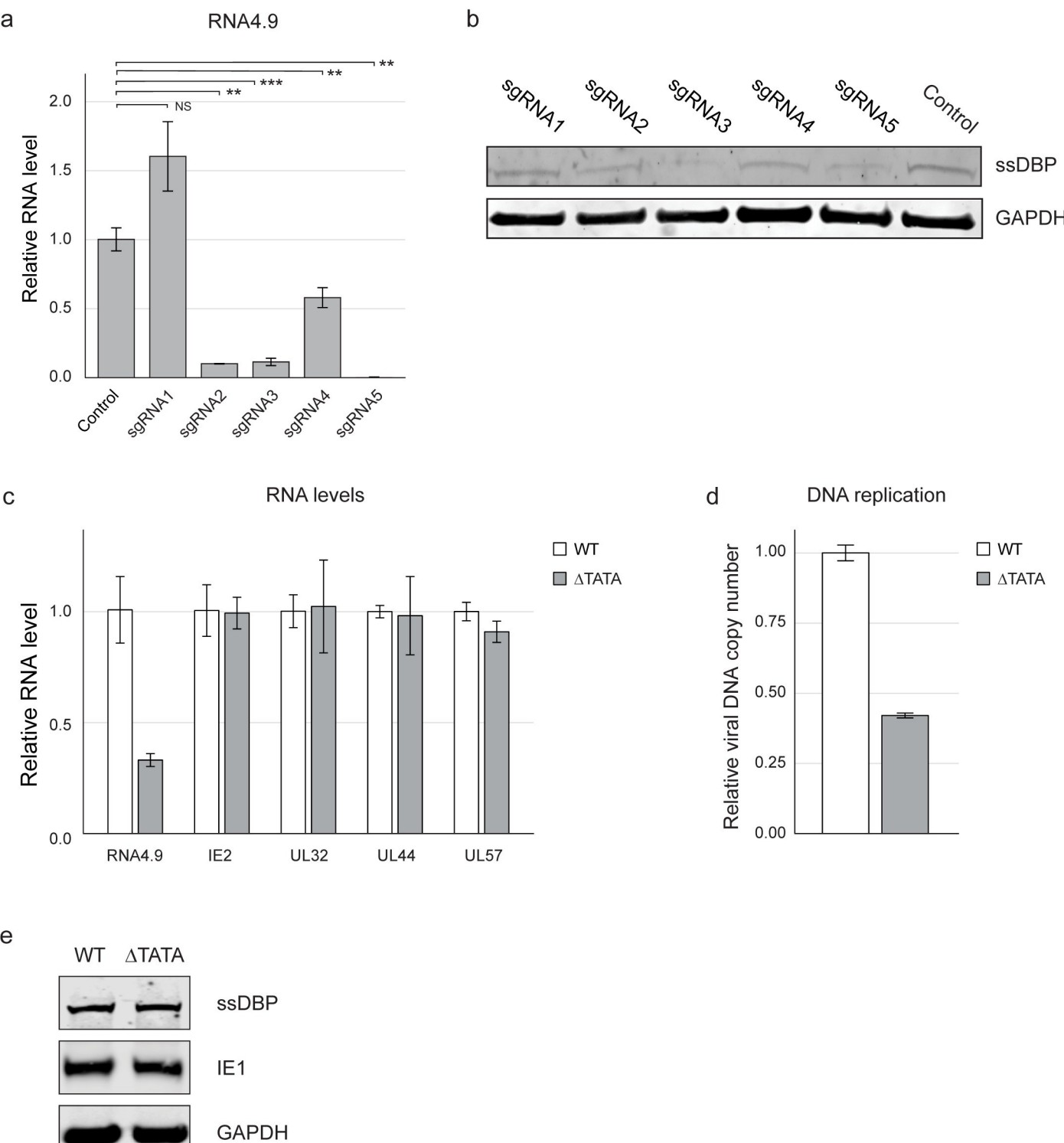

**Fig 6. ssDBP levels and viral DNA replication are strongly correlated with RNA4.9 transcription. a-b)** Fibroblasts expressing CAS9 and either a control sgRNA or one of five different sgRNAs targeting RNA4.9, as indicated, were infected with HCMV Merlin strain (MOI = 3), treated with PFA and harvested at 48 hpi. **a)** Relative RNA4.9 levels were quantified using RT-qPCR and normalized to the cellular ANXA5 transcript. **b)** ssDBP was detected by immunoblot analysis and GAPDH was used as a loading control. **c-e)** Fibroblasts infected with the ΔTATA or the parental WT Merlin strain (MOI = 1) were harvested at 48 hpi. **c)** Fibroblasts were treated with PFA during infection. Relative expression levels of indicated viral genes were quantified using RT-qPCR and normalized to the cellular ANXA5 transcript. **d)** Relative viral DNA levels were measured using qPCR using UL44 primers, and normalized to the cellular gene B2M. **e)** ssDBP, IE1/2 and the UL44 protein were detected by immunoblot analysis. Human GAPDH was used as loading control.

early and late gene expression did not seem to be affected (Fig 6C). The reduction in RNA4.9 transcription coincided with a two-fold reduction in viral DNA replication (Fig 6D), further supporting a role for RNA4.9 expression in regulating viral DNA replication. However, no major changes in ssDBP expression were detected (Fig 6E), implying that either a more substantial ablation of RNA4.9 is needed to affect ssDBP expression or that there is some effect of the use of CRISPR–Cas9 on expression of ssDBP.

To explore further the complex relationship between RNA4.9, *OriLyt* activity and UL57 expression, we used a previously generated HCMV mutant (OriShift; Fig 7A) in which *oriLyt* had been moved [27]. This relocation split the RNA4.9 gene, leaving 3052 bp of the 3' portion at the original site. As expected, the 3' portion of RNA4.9 was not expressed in cells infected with this mutant, whereas expression of the 5' portion relocated with *oriLyt* was partially reduced in the presence of PFA (Fig 7B). OriShift did not exhibit a major defect in IE1 expression (Fig 7C) in comparison with its parental WT virus, but did show a significant reduction in viral DNA synthesis (Fig 7D). Relocation of *oriLyt* and the 5' region of RNA4.9 resulted in a reduction of ssDBP levels in the presence of PFA (Fig 7C), but there were no significant changes in UL57 transcript levels (Fig 7E). By generating an HCMV mutant in which the 3' portion of RNA4.9 had been deleted (Δ3'), we demonstrated that the reduction in ssDBP levels in OriShift is not due to lack of expression of the 3' portion of RNA4.9 (S7A and S7B Fig). These results show that the interference with *oriLyt* activity caused by its relocation, independent of CRISPR-Cas9 cutting, affects ssDBP expression and that this effect is independent of the 3' region of RNA4.9.

## Overexpression of ssDBP rescues the defect caused by interference with *OriLyt* activity

In light of the drastic reduction observed in ssDBP expression and the essential role of ssDBP in viral DNA replication [45], we investigated whether overexpression of ssDBP could relieve the inhibition of viral DNA replication and growth of the OriShift mutant and in cells in which RNA4.9 promoter was targeted by CRISPR-Cas9. Fibroblasts were transduced with a lentivirus vector encoding ssDBP or mCherry. RNA4.9 KD in these fibroblasts was confirmed (S7C Fig). Significantly, overexpression of ssDBP increased viral DNA replication in cells infected by OriShift or its parental WT virus (S7D Fig), as well as in infected RNA4.9 KD and control cells (S7E Fig), suggesting that the ssDBP may be a limiting factor in viral DNA replication regardless of the interference with *oriLyt*. Overexpression of ssDBP partially rescued the viral growth defect in OriShift (Fig 7F) and in RNA4.9-Cas9 KD (Fig 7G), but did not increase viral titers in cells infected with WT virus or control KD. Overall, these results indicate that interference with RNA4.9 expression leads to a replication defect that is further enhanced by limiting levels of ssDBP, and that overexpression of ssDBP elevates viral DNA replication, which can compensate for diverse defects in *oriLyt* activity.

## MCMV encodes an RNA4.9 counterpart that also plays a role in viral DNA replication

Given that *OriLyt* is located upstream of the gene encoding ssDBP in all betaherpesviruses [32,46,47], and the potential role we discovered for RNA4.9 in regulating viral DNA replication, we investigated whether an RNA4.9 counterpart exists in MCMV. RNA-seq data from MCMV-infected mouse embryonic fibroblasts (MEFs) revealed a corresponding 1.6 kb transcript that is abundant, *oriLyt*-embedded, G+C-rich and lacking ORFs predicted to encode functional proteins, which we named RNA1.6 (Fig 8A). In contrast to RNA4.9, which is expressed prior to viral DNA replication, RNA1.6 seemed to be expressed with delayed late

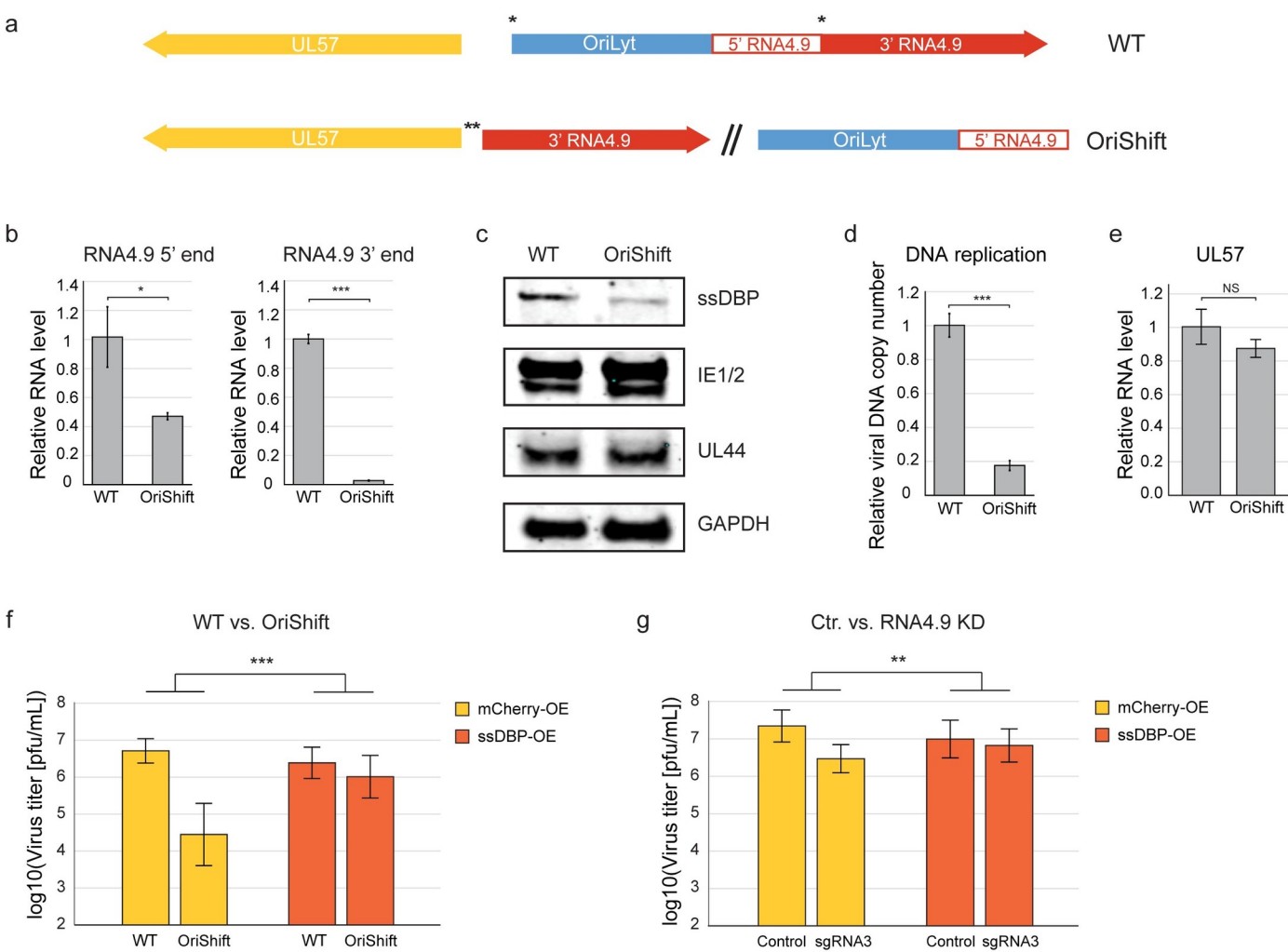

**Fig 7. ssDBP overexpression rescues the defect caused by interference with RNA4.9 expression. a)** Schematic representation of the OriShift mutant [27]. Asterisks mark the genomic location of the shifted *oriLyt* region. **b, c and e)** Fibroblasts were infected with OriShift mutant or the parental AD169 WT virus (MOI = 1), treated with PFA, and harvested at 48 hpi. **b)** Relative RNA4.9 5' and 3' transcript levels were quantified using RT-qPCR and normalized to the cellular ANXA5 transcript. **c)** ssDBP, IE1/2 and the UL44 protein were detected by immunoblot analysis. Human GAPDH was used as loading control. **d)** Fibroblasts were infected with OriShift mutant or the parental AD169 WT virus (MOI = 1) and harvested at 48 hpi, and relative viral DNA levels were quantified by qPCR using UL44 primers and normalized to the cellular gene B2M. **e)** Relative UL57 transcript levels were quantified using RT-qPCR and normalized to the ANXA5 cellular transcript. **f—g)** Fibroblasts overexpressing mCherry as control (yellow) or ssDBP (orange) were either infected with the OriShift mutant or the parental WT virus (MOI = 1) (f) or transduced with CAS9 and either a control sgRNA or a sgRNA targeting RNA4.9 (sgRNA3) and infected with HCMV Merlin strain (MOI = 1) (g). Viral titers were measured at 5 dpi by TCID50. **b, d and e)** Values and error bars represent the average and SD of triplicates. **b-g)** A representative analysis of at least two independent experiments is shown. **b, d and e)** two-sided *t*-test, as well as **f-g)** two-way ANOVA was applied (***p-value<0.001, **p-value<0.01, *p-value<0.05, NS, not significant).

kinetics (S8A Fig). The use of sgRNAs targeting RNA1.6 TSS reduced the level of RNA1.6 and inhibited MCMV DNA replication (Fig 8B and 8C). Since inhibition of viral DNA replication completely ablated RNA1.6 expression (S8B Fig), it was not possible to test whether RNA1.6 reduction was secondary to inhibition of viral DNA replication. We next tested whether expression of gene M57, encoding MCMV ssDBP, is affected by sgRNAs targeting RNA1.6 TSS sites. To exclude indirect effects due to inhibition of viral DNA replication, we performed the experiments in the presence of PFA. All three sgRNAs tested caused a reduction in ssDBP levels (Fig 8D) but no substantial effect on M57 transcript levels (Fig 8E). These results suggest that the coupling of ssDBP expression with *oriLyt* activity and a role for *oriLyt*-associated

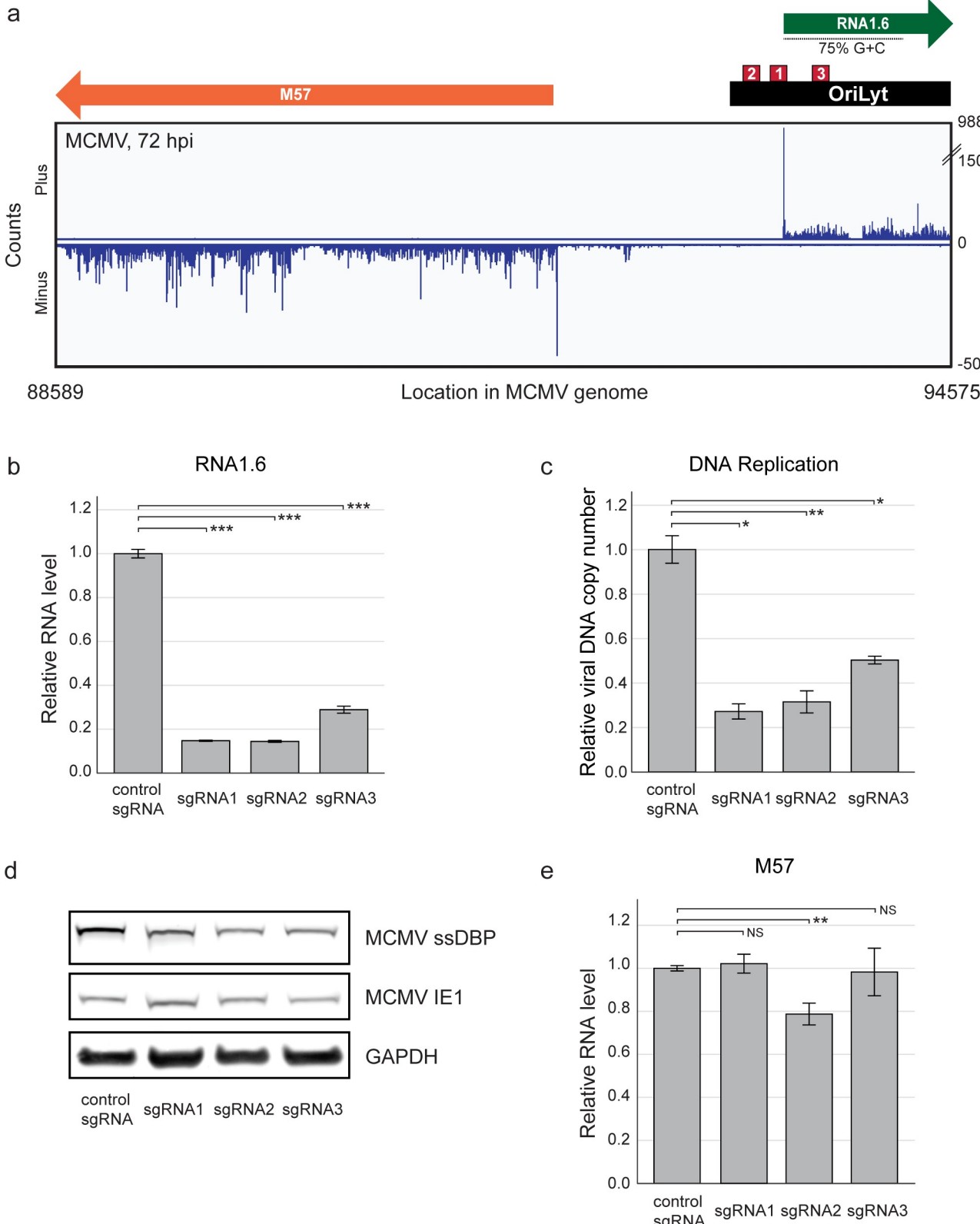

**Fig 8. MCMV RNA1.6 KD is associated with reduced ssDBP levels and inhibition of viral DNA replication. a)** Polyadenylated RNA profile of the MCMV Smith strain (NC_004065.1) genomic locus encompassing M57 (encoding ssDBP, orange), *oriLyt* (black) and RNA1.6 (green) with its G+C-rich region (dashed line), at 72 hpi (MOI = 1). sgRNA PAM sites (shown in red boxes; sgRNA1–93412, sgRNA2–93378, sgRNA3–93495) are displayed.

**b-e)** MEFs expressing CAS9 and either a control sgRNA or one of three different sgRNAs targeting the RNA1.6 TSS were infected with MCMV Smith strain (MOI = 1), left untreated (**b,c**) or treated with PFA (**d,e**) and harvested at 20 hpi. **b)** Relative RNA1.6 levels were quantified by RT-qPCR. RNA levels were normalized to mouse GAPDH. **c)** Relative viral DNA levels were quantified by qPCR using primers for gene M54. The murine gene Ino80 was used for normalization. **d)** MCMV ssDBP levels were detected by immunoblot analysis. Mouse GAPDH was used as loading control. **e)** Relative M57 transcript levels were quantified by RT-qPCR. RNA levels were normalized to the mouse transcript GAPDH. **b,c,e)** Values and error bars represent the average and SD of triplicates. A representative analysis of two independent experiments is shown. Two-sided *t*-test was applied (***p-value<0.001, **p-value<0.01, *p-value<0.05, NS, not significant).

lncRNA in DNA replication may be conserved in MCMV and might represent a conserved feature among betaherpesviruses.

## Discussion

The functions of HCMV lncRNAs remain largely unknown despite their high abundance during lytic infection. We have shown that, unlike the other three HCMV lncRNAs, RNA4.9 is localized to viral DNA replication sites in the nucleus, and that it is important for efficient viral DNA replication and growth. Furthermore, the use of RNA4.9-Cas9 KD caused a decrease in viral DNA replication and a substantial and specific reduction in ssDBP protein level. Although we were not able to connect unambiguously this reduction in ssDBP levels and the interference with RNA4.9 transcription, our results nonetheless provide strong evidence that ssDBP level is coupled to *oriLyt* activity.

Our findings indicate strongly that RNA4.9 is directly involved in viral DNA replication. First, RNA4.9 is transcribed early in infection and accumulates throughout infection within the viral DNA replication compartment. Second, RNA4.9 is transcribed within *oriLyt* and generates DNA-RNA base pairing, a feature that has been shown to be important for unwinding *oriLyt* in other herpesviruses. Third, RNA4.9-Cas9 KD resulted in substantial and specific reduction of expression of RNA4.9 that coincided with the reduction in viral DNA replication. Fourth, RNA4.9 KD using CRISPRi, in which RNA4.9 transcription is blocked but mutations are not generated, also resulted in a reduction in viral DNA replication. Fifth, a viral mutant lacking the RNA4.9 TATA box and producing reduced levels of RNA4.9 showed a concomitant reduction in viral DNA replication. Together, these findings indicate a direct involvement of RNA4.9 in viral DNA replication. However, it is important to note that, since these experiments involved targeting RNA4.9 transcription, it is not possible to distinguish between the act of transcription and the function of the RNA itself. Indeed, it has been suggested previously that initiation of replication at HCMV *oriLyt* involves transcriptional activation [48], indicating that the act of transcription might be the more important. Finally, in support of the possibility that RNA4.9 transcription *per se* plays a functional role (possibly through the formation of an R-loop), it was recently shown that there is an unusually high level of RNA polymerase II occupancy downstream of the RNA4.9 promoter [49].

HCMV *oriLyt* is a complex locus that has an asymmetric base distribution and contains several inverted and direct repeats [28]. The 5' part (1900 bp) of the RNA4.9 gene includes the G+C-rich region and is embedded in *oriLyt*. This makes it difficult to distinguish between changes in this region that interfere with RNA4.9 transcription and its downstream effects, and changes that interfere directly with *OriLyt* activity. Accordingly, we cannot rule out the possibility that DNA cleavage caused by the Cas9 system affected ssDBP expression and viral DNA replication, but previous work [50,51] and our data show that cleavage of HCMV DNA has only a modest impact on the expression of genes proximal to the cleavage site and minimal impact on HCMV replication. Moreover, the observation that much of the growth defect caused by RNA4.9-Cas9 KD could be rescued by overexpression of ssDBP further indicates that there is no major damage to *OriLyt*.

Our results indicate that ssDBP levels are tightly coupled to *oriLyt* activity; Following either CRISPR–Cas9 targeting or genetic relocation of *oriLyt*, ssDBP levels were substantially reduced. In addition, ssDBP could be detected during infection only when viral DNA replication had initiated and only in proximity to the replication compartment. Furthermore, ectopically expressed ssDBP was detected only at time points at which viral DNA replication had started, despite the transgene being expressed under an exogenous promoter. Although these results show that our ability to detect UL57 expression is coupled to viral DNA replication, treating cells with PFA did not lead to substantial reduction in ectopically expressed ssDBP. Furthermore, PFA treatment led to a reduction in endogenously expressed ssDBP that was comparable to the reduction in other viral early proteins (Fig 4D). Since PFA blocks viral DNA replication by inhibiting viral DNA synthesis, these results indicate that the initial steps of replication preceding DNA synthesis, such as *oriLyt* unwinding, are coupled to ssDBP levels rather than viral DNA synthesis itself.

Abundant, G+C-rich, *oriLyt*-embedded transcripts have been shown to be necessary for viral DNA replication in KSHV [38,52] and EBV [37,53,54]. Our results suggest that RNA4.9 forms an RNA-DNA hybrid through its G+C-rich 5' region, which may be important for the unwinding of *oriLyt*. It was indicated previously that two short viral RNAs (estimated by Northern blotting to be 300 and 500 nucleotides long) generate RNA-DNA hybrid regions within *oriLyt* [39]. The sequences encoding these putative RNAs were mapped to the G+C-rich 5' region of RNA4.9 in the opposite orientation relative to RNA4.9. RNA-seq of polyadenylated RNAs did not capture these RNAs [19,22], but small RNA sequencing did detect short RNA fragments that originate from this region [55], indicating that these RNAs, if they are indeed expressed, are not polyadenylated. In light of these data, it remains possible that the transcription pattern around the *oriLyt*, includes additional transcripts, which might be involved in its activity.

Our results support a model in which RNA4.9 transcription enhances the unwinding of *oriLyt* in a process that may involve R-loop formation. ssDBP accumulates concomitantly with *oriLyt* unwinding while perturbation of *oriLy* functions results in reduction in ssDBP expression probably due to its destabilization (Fig 9). In herpes simplex virus type 1 (an alphaherpesvirus) ssDBP has been reported to promote the formation of, and bind to, an R-loop *in vitro* [56]. Thus, it is possible that ssDBP is supporting the R-loop formation, resulting in a positive feedback loop that may promote viral DNA replication.

Expression of the RNA4.9 counterpart (RNA1.6) in MCMV also appeared to be linked to MCMV DNA replication and ssDBP expression. In addition, our recent transcriptome analysis of HHV6A and HHV6B revealed that these viruses also express lncRNAs that are transcribed from the viral origin of replication in the same orientation as RNA4.9 [57]. These observations indicate that the connection between the ssDBP, the *oriLyt* activity and lncRNA expression might reflect a conserved mechanism that facilitate tight regulation of viral DNA replication in betaherpesviruses.

Expression of ssDBP is essential for HCMV DNA replication [45] and viral growth [50]. Our results show that ssDBP levels are tightly coupled to *oriLyt* activity and that overexpression of ssDBP was capable of partially relieving the viral growth defect caused by RNA4.9-Cas9 KD or by relocation of *oriLyt*. Overexpression of ssDBP also significantly increased the level of viral DNA in WT HCMV-infected cells, suggesting that ssDBP may be a limiting factor in viral DNA replication. Interestingly, ssDBP overexpression did not increase viral titers in WT HCMV-infected cells, implying that the surplus of viral DNA does not translate into virus production. Thus, efficient viral production may require fine-tuning of the levels of ssDBP and genome copies, generating the most effective balance between them and the other resources required.

**Fig 9. Proposed model for the regulation of *oriLyt* activity.** Transcription of RNA4.9 regulates the activity of oriLyt, possibly through R-loop formation at the G+C-rich 5' end of RNA4.9, which may initiate the oriLyt unwinding. The levels of the ssDBP are coupled to the activity of *oriLyt* probably through stabilization of the protein on an unwound origin. Figure was made using BioRender.

Our study focused on a particular aspect of RNA4.9 function, and it is likely that this RNA has several roles operating in *cis* and *trans* during infection. For example, it has been suggested that RNA4.9 interacts with the UL84 protein and cellular components of polycomb repression complex 2 (PRC2) to promote transcriptional repression of major immediate-early promoter during HCMV latency [20]. In addition, several lines of evidence indicate additional and possibly independent roles for RNA4.9. First, RNA4.9 molecules accumulate to high levels early during the infectious cycle and appear quickly to outnumber the incoming and replicating genome molecules. Thus, the majority of RNA4.9 molecules are unlikely to be involved directly in R-loop formation. This is further supported by the finding that the majority of RNA4.9 molecules fill the interior of the replication compartment (Fig 1C), whereas nascent DNA synthesis occurs at the periphery of these sites (Fig 1D and [26,42]). In addition, RNA4.9 molecules were detected in juxtanuclear sites resembling the assembly compartment, whereas RNA1.2, RNA2.7 and RNA5.0 appeared to be excluded (Fig 1B and 1C), suggesting that RNA4.9 may also be involved in virion assembly. Finally, the length of RNA4.9 (4.9 kb) and the fact that the region encoding the 3000 bp at the 3' end is located outside essential regions of *oriLyt* point to functions that are independent of RNA4.9 transcription and R-loop formation. Some cellular lncRNAs are known to have structural roles in the nucleus. For example, the *NEAT1* lncRNA has been shown to serve as a platform for recruiting proteins to assemble paraspeckles [58,59]. Also, a nuclear structural role has been suggested recently for two gammaherpesvirus lncRNAs [60,61]. Whether RNA4.9 plays a structural role in the formation of replication compartments is an intriguing possibility that is worthy of further study.

In summary, our study indicates that the level of ssDBP is coupled to *oriLyt* activity during HCMV infection, and supports a role for an abundant viral nuclear lncRNA in the complex regulation of viral DNA replication. The fundamental nature of the mechanisms involved is likely reflected in the existence of counterparts of RNA4.9 in other betaherpesviruses.

## Material and methods

Human primary foreskin fibroblast cells and murine embryonic fibroblast (MEF) cells were grown at 37˚C in 5% (vol/vol) $CO_2$, in Dulbecco's modified Eagle's medium (DMEM, Biological Industries) supplemented with 10% (vol/vol) heat-inactivated fetal bovine serum (FBS, Life Technologies), 2 mM L-glutamine (Biological Industries), 0.1 mg/mL streptomycin and 100 U/mL penicillin (Biological Industries). The bacterial artificial chromosome (BAC)-derived AD169-GFP (pHG-1) virus and a mutant virus in which *oriLyt* had been moved to a different genomic location (pHG-6, OriShift) were kindly provided by M. Messerle at Hannover Medical School, Germany [27]. The complete genome sequences of these viruses were determined by Illumina sequencing. HCMV strain Merlin was used in all other experiments unless stated otherwise. Multiplicity of infection (MOI; plaque-forming units per cell) values used in HCMV infections are indicated in the Figure legends. MCMV strain Smith-GFP has been described previously [62].

### Construction of HCMV RNA4.9 ΔTATA and Δ3'mutants

These mutants were constructed from an HCMV strain Merlin BAC (pAL1111 [63]) by using the recombineering techniques described previously [63–66]. Briefly, a selectable *KanR/RpsL/lacZ* cassette flanked by appropriate HCMV sequences was transfected into *Escherichia coli* SW102 cells containing the parental BAC. Clones in which the cassette had recombined homologously into the BAC were selected positively using kanamycin. The inserted cassette was then replaced by the original sequence containing the desired mutation, and clones were selected negatively using streptomycin. BAC DNA was extracted using a Nucleobond BAC 100 kit (Macherey-Nagel) according to the manufacturer's instructions, and virus was reconstituted by transfection into human fibroblast cells using an Amaxa Basic Nucleofector kit for Primary Mammlian Fibroblasts (Lonza). The complete genome sequences of the viruses were determined by Illumina sequencing in order to ensure that the intended mutations were present and no others. Nucleotides 94297–94302 and 96114–99118 were absent from the ΔTATA and the Δ3' mutants, respectively (coordinates from GenBank accession no. AY446894.2).

### CRISPR plasmids

Oligonucleotides specifying 10 single guide RNAs (sgRNAs) mapping within the TATA box region of the HCMV strain Merlin RNA4.9 gene (94,208–94,436, AY446894.2) and three sgRNAs mapping within the TATA box region of the MCMV strain Smith RNA1.6 gene (93328–93554, U68299.1) were designed using benchling (https://benchling.com/) and cloned into the lentiviral vector lentiCRISPR v2 (Addgene#52961) [67]. Using Gibson cloning [68], a derivative of lentiCRISPR v2 was created in which Cas9 was replaced by a nuclease-inactive dCas9 gene fused to the KRAB repressor domain. dCas-KRAB was amplified from pHR-SFFV-KRAB-dCas9-P2A-mCherry [69]. Lentiviral particles were generated by co-transfection of the lenti-CRISPR v2 constructs and second-generation packaging plasmids (psPAX2, Addgene#12260; pMD2.G, Addgene#12259), using the jetPEI DNA transfection reagent (Polyplus-transfection)

with 293T cells according to the manufacturer's instructions. At 48 h post-transfection, supernatants were collected and passed through a 0.45 μm membrane (Millipore).

Cells grown in six-well plates were transduced with lentiviral vector particles in the presence of polybrene (8.33 μg/mL). At 48 h post-transduction, the cells were selected using 1.75 μg/mL puromycin. After 2–3 d of selection, the medium was replaced with fresh DMEM, and the selected cells were infected the next day. RNA4.9 expression was assessed by RT-qPCR. In subsequent experiments, sgRNA3 and sgRNA5 were used with Cas9 to knock down RNA4.9 levels. In the experiments applying dCas9-KRAB, sgRNA3 and sgRNA9 were used. Two different control sgRNAs were employed for HCMV: Control 1 targeting HCMV *oriLyt* without affecting RNA4.9 expression was used in the experiments presented in Figs 2 and S2; Control 2 targeting HCMV gene US2 [67] was used in the remaining experiments. A sgRNA (Control 3) targeting the non-essential MCMV gene m155 was used in the experiments presented in Fig 8. Sequences of sgRNAs are listed in Table 1.

**Table 1. Sequences and genomic locations of the single-guide RNAs used.**

| sgRNA | Sequence | Location in HCMV, Merlin |
|---|---|---|
| 4.9 1 | GCGGGCACGCCGGGTTTTAT | 94298-94317 |
| 4.9 2 | CTCTGAAAACCTATAAAACC | 94286-94305 |
| 4.9 3 | GGGCTCGCGCTCCCTAGGTG | 94208-94227 |
| 4.9 4 | AATTACCGCTCCGCCCACCT | 94257-94275 |
| 4.9 5 | AACCCTGCCGCGGACTGCGC | 94327-94346 |
| 4.9 9 | GCGGGAGCGGGCGCAGCGTG | 94366-94385 |
| Control 1 | CGGGTTTTATAGGTTTTCAG | 94288-94307 |
| Control 2 | GTCGGTTCGTCTTCGATCCG | 199703-199722 |
| sgRNA | Sequence | Location in MCMV, Smith |
| 1.6 1 | TATCCGCCACGATGACGCAT | 93413-93432 |
| 1.6 2 | AAATGGGCGCGGTTTCGCGG | 93379-93398 |
| 1.6 3 | TCCGAGGCGGCGGTCCGGAG | 93496-93515 |
| Control 3 | GACAACGACGATTACTGCGA | 215467-215489 |

## TCID$_{50}$ assay

$10^4$ human fibroblasts per well were plated in 96-well plates and infected with 10-fold serial dilutions of media collected from infected cells at 3 and 5 dpi, respectively. After 14 days, the dilutions showing a cytopathic effect were evaluated by light microscopy. TCID50 values per mL were calculated using the Spearman-Kaerber method [70].

## Immunofluorescence, EdU staining and FISH

Cells were plated on μ-Slide 8 well chambers (ibidi Gmbh) infected as indicated in the figure legends, washed once with PBS and fixed with 3.7% paraformaldehyde in PBS for 10 min at room temperature (RT). After fixation, the cells were washed twice with PBS and permeabilized with 0.5% (v/v) Triton X-100 in PBS for 20 min.

For metabolic labeling of nascent DNA, EdU (Jena Bioscience GmbH) was added to the culture medium at 10 μM and incubated for 30 min prior to fixation. After fixation and permeablization as described above, the cells were washed twice with PBS and rinsed once with TBS buffer (50 mM Tris-HCl, pH 7.5 and 150 mM NaCl). Azide-substituted 6-FAM

fluorophore was conjugated to incorporated EdU using a copper-catalyzed "click" chemistry as described previously [25]. Briefly, cells were incubated for 30 min at RT in conjugation solution (100 mM Tris-HCl pH 8.5, 10 μM fluorescent azide, 1 mM CuSO4 and 100 mM ascorbic acid, the last of which was made freshly and added last) and protected from light. After staining, the cells were washed three times with TBS containing 0.5% (v/v) Triton X-100 and then an additional three times with TBS (without detergent). FISH and IF were performed after EdU staining.

FISH samples were permeabilized with 70% (v/v) ethanol overnight at 4°C. The cells were incubated with FISH wash buffer (10% (v/v) formamide in 2xSSC (0.3 M NaCl, 30 mM sodium citrate, pH 7.0) for 5 min at RT. This was followed by overnight hybridization with FISH hybridization buffer (100 mg/mL dextran sulfate, 10% (v/v) formamide in 2xSSC) containing either 62.5 nM RNA4.9, 62.5 nM RNA2.7, 125 nM RNA1.2 or 125 nM RNA5.0 probes (Stellaris RNA FISH probes, Biosearch Technologies; S2 Table) at 37°C in a humidified incubator and protected from light. The samples were washed twice with warm wash buffer (10% (v/v) formamide in 2xSSC) at 37°C for 30 min followed by a wash with 2xSSC. The nuclei were stained with 1 μg/mL Hoechst in 2xSSC for 5 min at RT, and this was followed by a single wash with 2xSSC. To confirm the specificity of FISH to RNA rather than DNA, fixed and permeabilized cells were incubated prior to FISH with either RNase A (100 μg/mL in 2xSSC) or Turbo DNase I (20 U/mL in DNase reaction buffer, Thermo Fisher Scientific) for 100 min at 37°C and subsequently washed twice with warm (37°C) FISH wash buffer containing 15% (v/v) formamide. FISH probes for RNA5.0 covered the full gene sequence (both exons and the large intron). Detection of the UL44 protein and ssDBP was performed by immunostaining with anti-UL44 antibody (Virusys, CA006, 1:200 in PBS) for 2 h at RT or ssDBP antibody (Virusys, P1209) for 1.5 h at RT. The cells were washed 3 times with PBS and labeled with anti-mouse Alexa 647 conjugated secondary antibody (Jackson ImmunoResearch, 1:200 in PBS) for 1 h at RT. The cells were washed three times with PBS and counterstained with 1 μg/mL Hoechst. Anti-UL44 staining was performed after FISH against RNA4.9, whereas ssDBP detection required staining with primary and secondary antibodies prior to FISH, followed by post-fixation with 2% (w/v) paraformaldehyde in PBS for 10 min at RT. Imaging was performed on a Zeiss Axio Observer Z1 widefield microscope equipped with an X63 oil-immersion objective and an Axiocam 506 mono camera using ZEN imaging software (Zeiss).

## Subcellular fractionation

Fractionation of cytosolic and nuclear extracts was performed using the NER-PER Nuclear and Cytoplasmic Extraction Reagents Kit (ThermoFisher) according to the manufacturer's instructions, with the exception that 2 μl SUPERase In (Invitrogen) were added to CER I and 1 μl SUPERase In (Invitrogen) were added to CER II. RNA from these fractions was isolated using Trizol (Sigma-Aldrich) according to the manufacturer's instructions, and was subsequently used for RT-qPCR.

## Inhibitors

To inhibit viral DNA synthesis, 400 μg/mL of sodium phosphonoformate tribasic hexahydrate (PFA, Sigma-Aldrich) was added to culture medium after the viral inoculum had been removed by washing.

## Real-time PCR

Total cell RNA was extracted and purified using a Quick-RNA MiniPrep kit (Zymo Research), and cDNA was prepared from the RNA using a qScript cDNA synthesis kit (Quantabio) according to the manufacturer's instructions. Total cell DNA was extracted and purified using the QIAamp DNA Blood Mini Kit (Qiagen) according to the manufacturer's instructions. Quantitative PCR was performed using SYBR Green PCR master-mix (ABI) on the StepOnePlus or QuantStudio 6 Flex real-time PCR systems (Life Technologies) with the primers listed below.

The human *ANXA5* and mouse *GAPDH* mRNAs were used to normalize HCMV and MCMV RNA levels, respectively. The human *B2M* and mouse *Ino80* host genes were used to normalize HCMV and MCMV DNA levels, respectively. The relative levels of HCMV and MCMV DNA were estimated by quantification of the HCMV UL44 or UL55 and MCMV M54 genes, respectively. Primer sequences are listed in Table 2.

**Table 2. Primer sequences used for real-time PCR.**

| Gene | Forward primer sequence | Reverse primer sequence |
|---|---|---|
| RNA4.9 a for DRIP | GGGCCTCTGAAAACCTATAAAACCC | ATGGTGCTCCAGGGCGGT |
| RNA4.9 5'/ RNA4.9 b for DRIP | GGTGACTTTCTCGACGGTTC | ACGCTCCTAGGCTCTCGAC |
| RNA4.9 3' | GTAAGACGGGCAAATACGGT | AGAGAACGATGGAGGACGAC |
| UL55 | TGGGCGAGGACAACGAA | TGAGGCTGGGAAGCTGACAT |
| UL123 | TCCCGCTTATCCTCAGGTACA | TGAGCCTTTCGAGGACATGAA |
| UL44 | AGCAAGGACCTGACCAAGTT | GCCGAGCTGAACTCCATATT |
| UL99 | GGGAGGATGACGATAACGAG | TGCCGCTACTACTGTCGTTT |
| ANXA5 | AGTCTGGTCCTGCTTCACCT | CAAGCCTTTCATAGCCTTCC |
| B2M | TGCTGTCTCCATGTTTGATGTATCT | TCTCTGCTCCCCACCTCTAAGT |
| UL57 | TGAACGCAGAAACGCAGGAG | GAAATCCGCCTCCACCGTGA |
| APOE | CCGGTGAGAAGCGCAGTCGG | CCCAAGCCCGACCCCGAGTA |
| SLC22A1 | ACTGTCGTGGTGAGTGAGAG | GGAACCTGTCTCTGTCAGCT |
| RNA1.2 | TGACAACGCCTTGTATAGCC | AGACTGTCGTGGTCGATGAA |
| RNA2.7 | TCCATGTTTCCATCCTTTCA | AATCAGCGTTGCAGTAGTCG |
| GAPDH | TGGTATCGTGGAAGGACTCA | CCAGTAGAGGCAGGGATGAT |
| UL32 | GGTTTCTGGCTCGTGGATGTCG | CACACAACACCGTCGTCCGATTAC |
| M54 | CAGAAAGAGGTCATGACGCG | GAAGGGGAAGTGGAAGACGA |
| M57 | ATCTTCAAGGAGCGGATCGT | CTTGTACTGGATCTTGCGCC |
| RNA1.6 | CTTCCCGGCTACCCTCCT | GAGGAGCCGGACAGGAAC |
| Murine GAPDH | TCAAGCTCATTTCCTGGTATGACA | TAGGGCCTCTCTTGCTCAGT |
| 18S | CTCAACACGGGAAACCTCAC | CGCTCCACCAACTAAGAACG |
| Murine Ino80 | GCACTTCCTGGTTTTGCTGT | CACTGACTGGCGTGTTCAGA |
| UL57-flag | GGTCTTCTTCTCGGCGAGT | CTTGTCGTCATCGTCTTTGTAGTC |
| mCherry | ACCGCCAAGCTGAAGGTGAC | GACCTCAGCGTCGTAGTGGC |

## siRNA and ASO transfections

Human fibroblasts were refreshed at 4–6 hour prior to transfection with Gibco Opti-MEM I reduced serum medium (ThermoFisher Scientific). siRNAs (20 nM) and ASOs (100 nM), respectively, were transfected into the cells using Lipofectamine RNAiMAX transfection

reagent (ThermoFisher Scientific) according to the manufacturer's instructions. On the following day, the cells were washed with complete DMEM and infected with HCMV as described in the figure legends.

siRNA and ASO sequences are listed in Table 3.

**Table 3. Sequences of siRNAs and ASOs.**

| Name | 5'- 3' Sequence | 3'- 5' Sequence |
|---|---|---|
| Control siRNA | CGUUAAUCGCGUAUAAUACGCGUAT | AUACGCGUAUUAUACGCGAUUAACGAC |
| siRNA RNA1.2 | AGAAUCUCAUGAACUAGUCAACCAA | UUGGUUGACUAGUUCAUGAGAUUCUGC |
| siRNA RNA4.9 I | UCUGAUUCUCUGAAGAAUCACCGTC | GACGGUGAUUCUUCAGAGAAUCAGAAA |
| siRNA RNA4.9 II | AUAUGAUGAACCAAGAAUAAAACTC | GAGUUUUAUUCUUGGUUCAUCAUAUAU |

| Name | Sequence | |
|---|---|---|
| Control ASO | mG*mG*mC*mG*mA*mU*A*G*C*A*G*G*A*G*A*A*G*T*mC*mU*mG* mA*mA*mG | |
| ASO RNA4.9 I | mC*mU*mA*mC*mG*T*G*G*T*A*A*G*A*G*T*mC*mU*mU*mG*mG | |
| ASO RNA4.9 II | mG*mA*mC*mU*mG*T*C*T*A*T*G*G*T*T*A*mU*mG*mC*mA*mA | |

## RNA sequencing

Control cells and cells in which RNA4.9 was targeted by CRISPR were infected for 24h in the presence of PFA, total cell RNA was isolated using Trizol (Sigma-Aldrich) according to the manufacturer's instructions. mRNA was enriched using a Dynabeads mRNA DIRECT purification kit (ThermoFisher Scientific) according to the manufacturer's instructions, and RNA-seq libraries were prepared using a NEBNext Ultra Directional RNA Library Prep kit for Illumina (New England Biolabs) according to the manufacturer's instructions. For RNA-seq of MCMV infected cells, MEFs were harvested 72hpi libraries enriched for the 5' ends of transcripts were prepared as described previously [71].

## Illumina sequencing and data analysis

Raw sequence reads were generated using a NextSeq500 (Illumina). Prior to alignment, linker and polyA sequences (if present) were removed from the ends of reads. Alignment of the reads to reference sequences was performed using Bowtie (allowing up to two mismatches). Reads aligned to ribosomal RNA were removed, and the remaining reads were aligned to the HCMV strain Merlin or MCMV strain Smith genome sequences (AY446894.2 or U68299.1, respectively). Still-unaligned reads were aligned to 200bp sequences that spanned splice junctions. Reads with unique alignments (S1 Table) were used to compute the total number of reads for each viral gene. Differential expression analysis was done using DESeq2 [72]. For expression levels of the viral lncRNAs, only exonic reads were taken into account.

## DNA-RNA immunoprecipitation (DRIP)

DRIP was performed as described previously [37] with adjustments. Cells were washed three times with PBS, resuspended in lysis buffer (20 mM Tris pH 8, 4 mM EDTA, 20 mM NaCl, 1% (w/v) SDS, supplemented with 0.2 U/mL SUPERase In RNase Inhibitor (AM2694, ThermoFisher Scientific) and 0.7 µg/µL proteinase K; $10^7$ cells/ 500 µL), and incubated at 37˚C for 18 h. DNA was extracted using phenol-chloroform and precipitated in 75% (v/v) ethanol, 0.3 M sodium acetate (pH 8) supplemented with 1 µL Glycoblue (ThermoFisher Scientific). DNA was treated with RNaseA (20 ng/µL) in Tris-EDTA buffer and sonicated using a Bioruptor Pico (Diagenode) to generate fragments of 500–800 bp.

8 µg of sonicated DNA was resuspended in immunoprecipitation (IP) buffer (10 mM Sodium phosphate pH 7, 140 mM NaCl and 0.05% (v/v) Triton X-100). Residual sonicated DNA was used as input control for qPCR analysis. Subsequently, either 5 µg of anti-RNA-DNA hybrid antibody (S9.6, MABE1095; MERCK) or the relevant isotype antibody (IgG2s, clone S43.10; Miltenyi Biotec) were added to the DNA and incubated for 18 h at 4˚C. 50 µL protein G-Dynabeads (Invitrogen) was added per sample and incubated for 2 h at 4˚C. The DNA was eluted by incubating the beads with 250 µL proteinase digestion buffer (50 mM Tris-HCl pH 8, 10 mM EDTA, 0.5% (w/v) SDS and 0.3 µg/µL proteinase K) at 50˚C for 3 h while shaking. The DNA was precipitated in 75% (v/v) ethanol and 0.2 M NaCl overnight at -20˚C and analyzed by qPCR.

## Immunoblot analysis

Cells were lysed using ice-cold RIPA buffer (150 mM NaCl, 1% Triton X-100, 0.5% Na deoxy-cholate, 50 mM Tris-HCl [pH 8] and 0.1% (w/v) SDS) supplemented with a protease inhibitor cocktail (Sigma-Aldrich) for 10 minutes at 4˚C. Lysates were cleared by centrifugation at 4˚C for 10 minutes at 20800 x g. Proteins were separated on Bolt 4–12% Bis-Tris Plus polyacryl-amide gels (ThermoFisher Scientific) and blotted onto nitrocellulose membranes. The membranes were blocked with Odyssey Blocking Buffer (Li-COR) mixed 1:1 with TBST (150mM NaCl, 50 mM Tris-HCl pH7.5 and 0.1% (v/v) Tween), and immunoblotted with primary antibodies (1:1000 in TBST, 5% BSA and 0.05% (w/v) NaN3) for 1 h at RT or overnight at 4˚C. This was followed by three washes with TBST. The membranes were probed with secondary antibodies (1:10000 in TBST and 5% (w/v) skimmed milk powder) for 1 h at RT and washed three times with TBST. Fluorescent signal was acquired using an Odyssey CLx (LI-COR) and quantification using ImageJ. The primary antibodies used were against the following: GAPDH (Cell Signaling, 2118S), HCMV ssDBP (Virusys, P1209), HCMV UL44 protein (Virusys, CA006-100), HCMV IE1 and IE2 (Abcam, ab53495), HCMV UL84 protein (Virusys, CA144-500), FLAG-M2 (Sigma-Aldrich, F3165), mCherry (Abcam, ab205402), MCMV IE1 and MCMV ssDBP (both were a kind gift from S. Jonjic). The secondary antibodies used were as follows: IRDye 680RD goat anti-rabbit (Li-COR LIC-92668071), IRDye 680RD goat anti-mouse (Li-COR, LIC-92668070) and goat anti-chicken IgY (H+L) conjugated to Alexa Fluor-647 (ThermoFisher Scientific, A-21449).

## Genome editing detection assay

Total cell DNA was extracted and purified at 48 hpi using the QIAamp DNA Blood Mini Kit (Qiagen) according to the manufacturers instructions. An ~800 bp gene RNA4.9 fragment, covering the target locations of all sgRNAs, was amplified using KAPA 2G Robust PCR kit (KAPABIOSYSTEMS; forward primer AGTGCGCATGCGTCGGTA, reverse primer ACCTACCGTCGTCGTCGG) and used with the Alt-R Genome Editing Detection Kit (IDT) according to the manufacturer's instructions.

## Cloning of overexpression plasmids

The UL57 gene was amplified from purified HCMV DNA using primers that contained an overlap with the lentiCRISPR v2 vector: forward primer ggaccggttctagagcgctgccaccATGAGCC ACGAGGAACTAACCGCG, reverse primer gtttgttgcgccggatccTTACAACCGGCTGCGTT TGGCC; lower case characters represent the vector-overlapping regions). The vector was cut using XbaI and BamHI (NEB), and the UL57 fragment was cloned into it using the Gibson assembly method [68].

The UL57 gene was amplified from the lentiCRISPR v2-derived plasmid (described above) using primers that contain XbaI or BamHI restriction sites and a flag sequence, adding it downstream to the UL57 sequence. The primers used were as follows: forward primer, gaccggttctagagATGAGCCACGAGGAACTAACC, reverse primer, aataggatccTTA*CTTGTC GTCATCGTCTTTGTAGTC*CAACCGGCTGCGTTTGG; lower case characters represent restriction sites, and italic characters represent the flag sequence). The vector and insert were cut using XbaI and BamHI (NEB) and the UL57-flag fragment was ligated into the vector overnight at 16˚C.

The mCherry gene was inserted similarly into the lentiCRISPR v2 vector using restriction sites for AgeI and BamHI.

Lentiviral particles were generated by co-transfection of the lentiCRISPR v2-derived plasmids and second-generation packaging plasmids (psPAX2, Addgene#12260; pMD2.G, Addgene#12259) into HEK 293T cells using jetPEI DNA transfection reagent (Polyplus-transfection) according to the manufacturer's instructions. 48 hours post transfection, supernatants were collected and filtered through a 0.45 μm membrane (Millipore). Cells grown in 6-well plates were transduced with lentiviral vector particles in the presence of polybrene (8.33 μg/mL), selected at 48 hours post-transduction using 1.75 μg/mL puromycin, and infected at 3 days post-transduction. When RNA4.9 KD was performed, transduction with CRISPR plasmids was done prior to transduction of overexpression plasmids. Overexpression of mCherry was confirmed using fluorescent microscopy, and overexpression of ssDBP was confirmed by immunoblot in each experiment.

## Supporting information

**S1 Fig. RNA4.9 signal is RNA-specific. a)** Mock and HCMV Merlin strain-infected fibroblasts at 48 hpi (MOI = 5) were stained using fluorescent probes (white) against indicated HCMV lncRNAs. **b)** RNA4.9 was detected by RNA-FISH using fluorescent probes (white) in HCMV Merlin strain-infected fibroblasts at 48 hpi (MOI = 5). Cells were untreated, pretreated with DNase I or RNase A as indicated. **a-b)** Nuclei were counterstained with Hoechst (blue). **c)** At 48 hpi (MOI = 3), infected fibroblasts were fractionated into cytosolic (CYTO) and nuclear (NUC) fractions. Relative RNA4.9 and RNA2.7 levels were quantified using RT-qPCR and normalized to the cytosolic fraction and the cellular transcript ANXA5. (TIF)

**S2 Fig. RNA4.9 KD using CRISPRi inhibits HCMV DNA replication and growth. a)** Fibroblasts transfected either with control siRNA or siRNAs targeting RNA1.2 and RNA4.9, respectively, were infected with HCMV Merlin strain (MOI = 1). Relative levels of RNA1.2 and RNA4.9 were quantified using RT-qPCR at 48 hpi, and normalized to the cellular transcript ANXA5. **b)** Fibroblasts transfected either with control ASOs or ASOs against RNA4.9, were infected with HCMV Merlin strain (MOI = 1). Relative levels of RNA1.2 and RNA4.9 were quantified using RT-qPCR at 48 hpi, and normalized to the cellular transcript ANXA5. **c-f)** Fibroblasts expressing dCAS9 and either a control sgRNA or one of two different sgRNAs targeting the RNA4.9 promoter (sgRNA3 and sgRNA9) were infected with HCMV Merlin strain (MOI = 0.1). **c)** Relative RNA4.9 levels were quantified using RT-qPCR at 48 hpi and normalized to the human transcript ANXA5. **d)** Relative viral DNA levels were quantified using qPCR at 48 hpi using UL55 primers, and normalized to the cellular gene B2M. **e)** Viral titers were measured 5 days post infection (dpi) by TCID50. **f)** Relative levels of the UL123 (IE1), UL44 and UL99 transcripts were quantified using RT-qPCR at 48 hpi and normalized to the cellular ANXA5 transcript. **c-f)** Values and error bars represent the average and SD of triplicates. A representative analysis of two independent experiments is shown. Two-sided *t*-test

was applied (\*\*\*p-value<0.001, \*\*p-value<0.01, \*p-value<0.05, NS, not significant).
(TIF)

**S3 Fig. RNA4.9-Cas9 KD does not directly affect viral gene expression.** Fibroblasts expressing CAS9 and either a control sgRNA or a sgRNA targeting the RNA4.9 promoter (sgRNA3) were infected with HCMV Merlin strain (MOI = 2) and treated with PFA. Relative levels of the indicated viral genes, including RNA4.9, were quantified using RT-qPCR at 48 hpi and normalized to the cellular transcript ANXA5.
(TIF)

**S4 Fig. RNA4.9-Cas9 KD does not affect UL84 expression and its recruitment to the viral replication compartment.** Fibroblasts expressing CAS9 and either a control sgRNA or a sgRNA targeting RNA4.9 (sgRNA3 or sgRNA5, as indicated) were infected with HCMV Merlin strain (MOI = 3). **a)** UL84 (red) was detected using IF at 48 hpi in the presence of PFA. **b)** Quantification of UL44 and UL84 protein levels from the immunoblot analysis in (Fig 4D), normalized to the levels of GAPDH.
(TIF)

**S5 Fig. ssDBP is stabilized by viral DNA replication. a)** Relative expression of UL57 transcript during HCMV infection as measured by RNA-seq [24]. **b)** Fibroblasts were infected with HCMV Merlin strain (MOI = 2) and harvested at the indicated time points post infection. Relative viral DNA levels were quantified using qPCR at the indicated time points, using UL44 primers and normalized to the cellular gene B2M.
(TIF)

**S6 Fig. Various sgRNAs lead to efficient Cas9 targeting.** Fibroblasts expressing CRISPR-Cas9 and five different sgRNAs targeting the RNA4.9 TSS region or a control sgRNA were infected with HCMV Merlin strain (MOI = 3), treated with PFA and harvested at 48 hpi. A T7 endonuclease I mismatch cleavage assay was conducted to estimate the genome editing efficiency of the RNA4.9 loci. The relative quantification of band intensities (which indicates a mutated sequence) is presented, normalized to the signal obtained using the control sgRNA.
(TIF)

**S7 Fig. ssDBP overexpression rescues the defect caused by interference with RNA4.9 expression. a-b)** Fibroblasts were infected with the Merlin Δ3' mutant or the parental strain (MOI = 1), treated with PFA, and harvested at 48 hpi. **a)** Relative RNA4.9 5' and 3' levels were quantified using RT-qPCR and normalized to the cellular ANXA5 transcript. **b)** ssDBP and IE1/2 were detected by immunoblot analysis. Human GAPDH was used as a loading control. **c-e)** Fibroblasts expressing mCherry as control (yellow) or ssDBP (orange) were infected, as indicated (MOI = 1), and harvested at 48 hpi. **c)** Relative RNA4.9 levels were quantified using RT-qPCR and normalized to the cellular ANXA5 transcript in fibroblasts expressing CRISPR-Cas9 and a sgRNA targeting the RNA4.9 TSS region (sgRNA3)or a control sgRNA and infected with the HCMV Merlin strain. **d)** Fibroblasts were infected with the OriShift mutant virus or the parental strain (AD169). Relative viral DNA levels were quantified by qPCR using UL44 primers, and normalized to the cellular gene B2M. **e)** Fibroblasts expressing CAS9 and either a control sgRNA or a sgRNA targeting RNA4.9 TSS region (sgRNA3) were infected with HCMV Merlin strain. Relative viral DNA levels were quantified using qPCR and UL44 primers, and normalized to the cellular gene B2M. **a, c-e)** Values and error bars represent the average and SD of triplicates. A representative analysis of two independent experiments is shown. Two-sided *t*-test was applied (\*\*\*p-value<0.001, \*\*p-value<0.01, n.d, not detected).
(TIF)

**S8 Fig. Kinetics of MCMV RNA1.6 expression. a)** Relative RNA1.6 transcript levels in MCMV Smith strain-infected MEFs (MOI = 1) were quantified by RT-qPCR at indicated time points post infection. RNA levels were normalized to the mouse GAPDH transcript. **b)** MEFs were either infected with MCMV Smith strain (MOI = 1) or left uninfected, and the infected cells were either treated or untreated with PFA. Relative RNA1.6 levels were quantified by RT-qPCR at 20 hpi. RNA levels were normalized to the mouse 18S rRNA. **a-b)** Values and error bars represent the average and SD of triplicates. A representative analysis of two independent experiments is shown.
(TIF)

**S1 Table. Differential expression analysis of viral transcripts in RNA4.9 KD vs. control cells.**
(XLSX)

**S2 Table. smFISH probes sequences.**
(XLSX)

# Acknowledgments

We thank Richard Stanton (Cardiff University) for providing the HCMV Merlin BAC and advising on the generation of the ΔTATA and Δ3' viruses, Martin Messerle and Eva Borst (Hannover Medical School) for providing us the OriShift mutant, Stipan Jonjić (University of Rijeka) for providing the MCMV IE1 and M57 antibodies, Dan Eliahu and Dan Hamershlak for technical assistance. N.S-G is an incumbent of the Skirball Career Development Chair in New Scientists.

# Author Contributions

**Conceptualization:** Julie Tai-Schmiedel, Sharon Karniely, Michal Schwartz, Noam Stern-Ginossar.

**Data curation:** Julie Tai-Schmiedel, Sharon Karniely, Adi Ezra, Aharon Nachshon, Nicolás Suárez.

**Formal analysis:** Julie Tai-Schmiedel, Sharon Karniely, Aharon Nachshon.

**Funding acquisition:** Noam Stern-Ginossar.

**Investigation:** Julie Tai-Schmiedel, Sharon Karniely, Adi Ezra, Erez Eliyahu.

**Methodology:** Julie Tai-Schmiedel, Sharon Karniely, Betty Lau, Karen Kerr, Nicolás Suárez, Noam Stern-Ginossar.

**Project administration:** Julie Tai-Schmiedel, Noam Stern-Ginossar.

**Resources:** Betty Lau, Karen Kerr, Nicolás Suárez, Andrew J. Davison, Noam Stern-Ginossar.

**Software:** Aharon Nachshon.

**Supervision:** Noam Stern-Ginossar.

**Validation:** Julie Tai-Schmiedel, Sharon Karniely.

**Visualization:** Julie Tai-Schmiedel, Sharon Karniely, Aharon Nachshon.

**Writing – original draft:** Julie Tai-Schmiedel, Sharon Karniely, Michal Schwartz, Andrew J. Davison, Noam Stern-Ginossar.

**Writing – review & editing:** Julie Tai-Schmiedel, Sharon Karniely, Michal Schwartz, Andrew J. Davison, Noam Stern-Ginossar.

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
