## [Decision Letter · Decision Letter 0]

18 Nov 2019

Dear Dr. Stern-Ginossar,

Thank you very much for submitting your manuscript "Human cytomegalovirus long noncoding RNA4.9 regulates viral DNA replication" (PPATHOGENS-D-19-01898) for review by PLOS Pathogens. Your manuscript was fully evaluated at the editorial level and by independent peer reviewers. The reviewers appreciated the attention to an important problem, but raised some substantial concerns about the manuscript as it currently stands. These issues must be addressed before we would be willing to consider a revised version of your study. We cannot, of course, promise publication at that time.

We therefore ask you to modify the manuscript according to the review recommendations before we can consider your manuscript for acceptance. Your revisions should address the specific points made by each reviewer.

(1) A letter containing a detailed list of your responses to the review comments and a description of the changes you have made in the manuscript. Please note while forming your response, if your article is accepted, you may have the opportunity to make the peer review history publicly available. The record will include editor decision letters (with reviews) and your responses to reviewer comments. If eligible, we will contact you to opt in or out.

(2) Two versions of the manuscript: one with either highlights or tracked changes denoting where the text has been changed; the other a clean version (uploaded as the manuscript file).

Additionally, to enhance the reproducibility of your results, PLOS recommends that you deposit your laboratory protocols in protocols.io, where a protocol can be assigned its own identifier (DOI) such that it can be cited independently in the future. For instructions see http://journals.plos.org/plospathogens/s/submission-guidelines#loc-materials-and-methods

We hope to receive your revised manuscript within 60 days. If you anticipate any delay in its return, we ask that you let us know the expected resubmission date by replying to this email. Revised manuscripts received beyond 60 days may require evaluation and peer review similar to that applied to newly submitted manuscripts.

[LINK]

Sincerely,

Eain A Murphy, Ph.D.

Associate Editor

PLOS Pathogens

Shou-Jiang Gao

Section Editor

PLOS Pathogens

Kasturi Haldar

Editor-in-Chief

PLOS Pathogens

orcid.org/0000-0001-5065-158X

Grant McFadden

Editor-in-Chief

PLOS Pathogens

orcid.org/0000-0002-2556-3526

Dear Naom,

I hope this finds you well. I have sent your manuscript to three experts in the Herpesvirus field who all have experience with Viral DNA replication. All three reviewers viewed your work as very important and appreciated the use of highly innovative tools to answer critical questions about HCMV replication. Their views were wide ranging on the acceptance of the manuscript and two reviewers (specifically #2 and #3) had significant issues with the interpretation and presentation of the data which undermined their support of the manuscript. I however find the manuscript of high value and am inclined to allow you to read the comments and see if you think they are addressable in a reasonable time frame. As such, I rendered a decision of Major Revision. If you are able to address the concerns of Reviewer #s 2 & 3, I would be inclined to send it back to them for review. While this is not the decision you were hoping for, do understand that if you are able to accommodate those two reviewers, I will be highly supportive of the work.

Cheers,

Eain Murphy

Reviewer's Responses to Questions

**Part I - Summary**

Reviewer #1: Upon characterizing the subcellular distribution of the four major lncRNAs encoded by HCMV, Tai-Schmiedel et al commence this study by showing that RNA1.2, RNA2.7 and RNA5.0 accumulate in the cytoplasm of infected cells while RNA 4.9 largely accumulates within nuclear viral replication compartments. Using Crispr-cas9 and CRISPRi methods, the authors create the first hypomorphic functional allele for RNA4.9 that maintains the critical integrity of oriLyt, elegantly overcoming a significant impediment that has prevented the investigation of RNA4.9 function during HCMV reproduction. They conclusively show that interfering with RNA4.9 accumulation directly inhibits viral DNA synthesis and reduces virus productive growth. Furthermore, effects of RNA4.9 on virus gene expression were shown to be secondary to effects on DNA synthesis. Using DNA-RNA hybrid immunoprecipitation, the authors provide evidence that GC rich regions specifically from the RNA4.9 RNA are enriched in these structures. Finally, they present evidence that RNA4.9 post-transcriptionally controls levels of the HCMV UL57 ssDBP and that ectopic expression of ssDBP restores replication of recombinant viruses containing RNA4.9 hypomorphic alleles. A similar RNA4.9 counterpart is identified in MCMV and shown to regulate viral DNA replication.

The work is significant as it is the first to establish biological function for the HCMV lncRNA4.9 in viral DNA synthesis and characterize the phenotype of RNA4.9 hypomorphic alleles in the productive growth cycle. It uses creative cutting edge genome editing methods to overcome a key obstacle limiting genetic analysis in large viral genomes that encode overlapping cis and trans-acting functions within discrete, compact regions and creates the first hypomorphic allele of lncRNA4.9 that doesn't incidentally damage orilyt. Besides this creative engineering strategy, the experiments are well executed, the data are rigorous and compelling and the manuscript is well written. The work is of broad interest to many investigators studying the many roles of non coding RNAs in infection biology and seeking to engineer mutations in overlapping genes without damaging critical cis-acting regulatory elements. I have no substantial, essential comments that would significantly improve this already strong manuscript.

Reviewer #2: The functions of the four long noncoding RNAs encoded by human cytomegalovirus are not well-characterized. This manuscript investigates the function of lncRNA4.9. The authors observe localization in the viral nuclear replication compartment during HCMV infection of fibroblasts. They apply a loss-of-function approach using gene editing in the lncRNA4.9 region. By this approach, they uncover an impact on viral DNA replication and the level of the viral ssDNA binding protein. A parallel approach in the MCMV system led to similar outcomes. The authors provide a model whereby the lncRNA4.9 binds the oriLyt to form an RNA-DNA hybrid that impacts the initiation of DNA replication and reduces ssDNA binding protein levels. However, the experimental approach used to edit the lncRNA4.9 will lead to dsDNA breaks that make the impact on oriLyt function and neighboring transcription very difficult to parse.

Reviewer #3: The manuscript by Tai-Schmiedel at al. “Human cytomegalovirus long noncoding RNA4.9 regulated viral DNA replication” is an impressive paper. Many of their experiments answer some of the basic questions regarding the expression and localization of the viral lncRNAs. This is an important area of research given the abundance of the viral lncRNA and how little is known about the functions during viral replication. The identification of a RNA4.9 homologue in MCMV is an interesting discovery and extends the work to a possible mechanism used by other beta herpesviruses. There are concerns regarding the approach to KD RNA4.9 and the interpretation of the role of RNA4.9 in DNA replication given its overlap with oriLyt, and how UL57 (ssDBP) is regulated by RNA4.9. These concerns damp some of the enthusiasm for the manuscript. Specific questions are listed below.

**Part II – Major Issues: Key Experiments Required for Acceptance**

Reviewer #1: none.

Reviewer #2: Major concern

1. Their approach to knock-down the gene of interest using WT Cas9 will lead to complex and varied mutations and possibly indirect host DNA damage responses.

The authors have used gene-editing with wt Cas9 to generate dsDNA breaks in the CMV genomes. This is not an appropriate way to transiently knock-down a specific gene of interest in the context of infection (why not use siRNA?). This type of approach is especially troublesome in the area of the genome that houses both the origin of lytic replication and the regulatory region of the DNA replication proteins that is focused on. A break in this area can cause non-specific host DNA damage responses, loss-of-function of the origin and disruption of the normal architecture of the genome that mediates transcription of genes in that area. Given the complexity of this region and the lack of characterization of the ensuing genome edits, it is impossible to interpret the downstream effects. The use of multiple gRNAs and seeing disparate effects merely indicates the efficiency of edits and damaged genomes differs. Their control gRNA will likely not cut at all- they might want to target an innocuous region (this is obviously hard to do). They do not sequence the edited genomes. Cas9 systems lead to a spectrum of mutations. Conclusions about the direct effect of loss of the lncRNA4.9 can not be made using this approach.

If the authors wish to use this method to edit the virus, they might purify and characterize the mutants prior to use in infection experiments to better ensure that only lncRNA4.9 is directly impacted. This will be very difficult to do. More refined mutations such as disruption of initiation or insertion of a transcriptional stop would seem much cleaner.

The use of a mutant Cas9 that impairs transcription initiation is a more reasonable approach and there is one supplemental figure demonstrating an impairment in lncRNA expression and replication. How does the alteration of transcription in this region impact the ‘firing’ of the lytic origins? Rescue by ectopic lncRNA4.9 expression should be demonstrated.

2. Experimental reproducibility and rigor throughout the paper.

In general, the MOI varies within each figure and across nearly all figures in the paper. This makes integration of RNA, DNA, Protein, and IFA data very difficult. Specific concerns with each figure are described below.

Figure 1.

1A appears to be previously published data. What was the cell type and MOI? How do these levels relate to coding transcripts typical of IE, E, L.

1B. What is the nuclear to cytoplasmic ratio? Why not show dual color as in 1C. Is there a software-based enumeration of this data so we can learn from hundreds of cells, not single cell staining?

1B-1C. Where is the mock control? MOI5.0 vs MOI1.0 in 1B. There is no direct demonstration of viral DNA replication compartments

Figure 2.

2A. There seem to be disparities in the impact on RNA, DNA, virus production between sgRNA3 and 5. The impact on gene expression seems to be greater with sgRNA3 at 24 hpi and with sgRNA5 at 48 hpi. Why was sgRNA 3 used moving forward?

Figure 3.

3A. This was based on one experiment. Why was a low MOI used for this? Why weren’t more genes impacted.

3B. The normalization of this data is really difficult to understand. There is no data regarding the oriLyt.

Figure 4.

A and b. 24 hpi is shown without PFA and the 48 hr timepoint is with PFA. This is a strange comparison. PFA abolishes viral replication compartments, does this change the nuclear localization of RNA4.9?

C. Isn’t it expected that PFA or anything that abrogates lytic replication will reduce viral gene expression?

D. no error bars

E. This should be provided with the exact coordinates of the sgRNAs used earlier in the manuscript.

Figure 5.

How many experimental replicates does this represent?

It is difficult to compare transcripts to protein. Such correlations are not very valuable and no viral DNA replication data is provided for reference. How is the quantitation of transcripts and proteins from an overexpressed gene informative? If PFA is used, what happens to ssDNADBP?

Figure 6.

Different sgRNAs in the lncRNA4.9 region have differential editing of the genome that leads to dsDNA breaks. This does not show a direct effect of one transcript to another.

Figure 7.

7D. This is an unusual mutant. Does the previous publication describe a replication defect at this MOI?

7F. Does ssDNA overexpression increase DNA copy number?

F and G. What was the statistical test used? The impact of ssDBP is very slight.

B-F, if you overexpress full-length lncRNA4.9, do you rescue ssDBP, replication and infectious virus production back to WT levels?

Figure 8.

A Where do the sgRNA bind and edit in MCMV? What are those edits?

3. The model is not supported by the data.

There are no experiments using proteasome inhibitors to support premise of ssDNABP instability. No data for direct interaction of lncRNA4.9 with the oriLyt. No data for impact on nascent DNA replication or unwinding at oriLyt.

Reviewer #3: The timing of the onset of DNA replication and the expression of RNA4.9 needs to be clarified. DNA replication begins around 24 hrs post infection. The expression kinetics of RNA4.9 needs to be confirmed at earlier time points before DNA replication begins in order to determine the role of RNA4.9 in DNA synthesis. The question is, what is the level of RNA4.9 between 24 and 36 hrs? The relationship between the levels of RNA4.9 and amount of newly synthesized DNA resulting in an increase in template which could be used to produce more RNA4.9. This brings into question if the results are really due to a decrease in RNA4.9 or if it is disrupting the cis-elements within oriLyt which would result in a decrease in DNA replication and then a decrease in RNA4.9. The other issue with the kinetics of RNA4.9 expression is that some of the experiments were performed at 24 hrs (Fig. 3 panel a) when they have not established that RNA4.9 is expressed at significant levels by 24 hpi. Fig. 1A extrapolates the levels of RNA4.9 from 24 to 72 hpi and it’s not clear if RNA4.9 is an early transcript or expressed after the onset of DNA synthesis. The knockdown of RNA4.9 (Fig. 2 panel a) is at 48 hpi, but that is already after the onset of DNA replication at 24-36 hpi.

In regards to the CRISPR experiments (Fig. 2, Fig. 4 and Fig. 6), what was the percentage of HCMV genomes that carried the sgRNA induced RNA4.9 promoter mutation? The authors attempt to answer this question with the relative induced mutations shown in Figure S6 but it doesn’t give an absolute value to how many genomes have the mutation since it is a relative comparison between the specific gRNA to the control gRNA.

This brings up another issue, the control elements within the origin and the RNA4.9 promoter. Promoters are often associated with activity of lytic origins, and the act of transcription is thought to help open the DNA for initiation of DNA synthesis or allow for assemble of the core enzymatic machinery. The SV40 promoter can functionally replace the HCMV oriLyt promoter, demonstrating that the act of transcription is an important process for oriLyt-dependent DNA replication. The RNA4.9 promoter may have additional functions beyond the expression of the RNA4.9 transcript, it may in fact be a control element within the origin for initiating. The authors have not ruled out the possibility that they are disrupting the function of the cis-elements in the origin. This is a very difficult issue to resolve but there is no evidence that the phenotype described is due to the disruption of the RNA4.9 promoter which causes a decrease in the expression of RNA4.9 or if the disruption of the RNA4.9 promoter is directly affecting the ability of the origin to function normally.

The R-loop within the origin has previously been described. In addition to the RNA4.9 R-loop there are other RNAs that have been identified from the origin. This was first reported in Prichard et al. J Virol. 1998 Sep;72(9):6997-7004 “Identification of persistent RNA-DNA hybrid structures within the origin of replication of human cytomegalovirus”. Interestingly, the 5’ region of RNA4.9 is within the transient plasmid-based replication assay and maybe functional in that system as well. The viral protein UL84 was also shown to interact with the stem-loop structure at the 5’ region of the RNA-DNA hybrid in Colletti et al. J Virol. 2007 Jul;81(13):7077-85 “Human Cytomegalovirus UL84 Interacts with an RNA stem-loop sequence found within the RNA/DNA hybrid region of oriLyt”.

Fig. 4 panel c and d. Based on the western blot it seems like there might also be a reduction in the amount of UL84 and UL44 protein in cells treated with the sgRNA3 and not a specific reduction in the ssDBP levels. The graph in panel d shows the ssDBP relative to GAPDH, what are the levels of UL84 and UL44 in the same graph. Additionally, with panel f what are the relative levels of early transcripts UL84 and UL44 compared to UL57?

Fig. 6 and 7. In figure 6 there is a decrease in RNA4.9 from the specific sgRNA and a decrease in UL57 RNA, while in Figure 7 there decrease in RNA4.9 in the OriShift mutant but the expression of UL57 is not significantly different in the WT and OriShift mutant (panel e). This difference in the amount of UL57 transcript detected (significantly different in cells where RNA4.9 is KD with Cas9 and sgRNA, but not significantly different in the oriShift mutant where RNA4.9 levels are reduced through genomic shifting) is not adequately discussion as to why there is an inconsistency in the method used to reduce RNA4.9 and the levels of UL57 that are reported.

Figure S2. The other compounding factor is if the CRISPRi protein, which binds to DNA to block transcription without the nuclease activity, could block the ability of the viral DNA initiation protein in a similar manner, and if that’s true then the results could be due to the blocking IE2 or UL84 at the origin in addition to blocking transcription.

**Part III – Minor Issues: Editorial and Data Presentation Modifications**

Reviewer #1: 1) It would be helpful to include a second guide RNA (ie:sgRNA5) in at least panel F of figure 4

2) Does PFA treatment limit accumulation of ectopically expressed ssDBP flag in HCMV infected cells? This would be predicted to occur and is simple to test.

Editorial comments to address in the text:

Line 199-200 revise to state,” We next tested the possibility that RNA4.9 KD affects ssDBP stability perhaps as an outcome of faulty unwinding of oriLyt. “ 

Line 205, replace “started” with “first observed”

Line 207-209: As written, this is somewhat confusing, whereas it is very clear in the figure 9 legend. I think the authors are combining too much in a single sentence and as a result may make it harder for the generalist reader to separate experimental findings from more interpretative speculation.

Figure 5 and S5 legends: the authors measure DNA accumulation, which of course reflects DNA replication. The term “initiation”, however, reflects a discrete phase of the replication process that is inferred by the authors, and while likely correct, has not been demonstrated experimentally. Perhaps the term “initiation” could be removed from the legend and retained in the text in the speculative manner in which it is already used by the authors.

Reviewer #2: (No Response)

Reviewer #3: Minor points:

Page 4 line 66-68: In regards to the reports of viral lncRNAs, Noriega et al. J Virol. 2014 Aug;88(16):9391-405 “Human cytomegalovirus modulates monocyte-mediated innate immune responses during short-term experimental latency in vitro” has also identified HCMV lncRNAs during latency.

PLOS authors have the option to publish the peer review history of their article (what does this mean?). If published, this will include your full peer review and any attached files.

Reviewer #1: No

Reviewer #2: No

Reviewer #3: No

---

## [Editor Report · Decision Letter 1]

7 Feb 2020

Dear Dr. Stern-Ginossar,

We are pleased to inform you that your manuscript 'Human cytomegalovirus long noncoding RNA4.9 regulates viral DNA replication' has been provisionally accepted for publication in PLOS Pathogens.

Before your manuscript can be formally accepted you will need to complete some formatting changes, which you will receive in a follow up email. A member of our team will be in touch within two working days with a set of requests.

Best regards,

Eain A Murphy, Ph.D.

Associate Editor

PLOS Pathogens

Shou-Jiang Gao

Section Editor

PLOS Pathogens

Kasturi Haldar

Editor-in-Chief

PLOS Pathogens

orcid.org/0000-0001-5065-158X

Michael Malim

Editor-in-Chief

PLOS Pathogens

orcid.org/0000-0002-7699-2064

Dr Stern-Ginossar,

I have read your comments to reviewers and re-read your manuscript and it is in my opinion that you have improved the manuscript a great deal, took the suggestions seriously and have performed a significant amount of work to make this manuscript acceptable for publication without further review. It represents a nice body of work and you and your co-authors should be proud. Please contact me if there is any way I can be of assistance. Again, congratulations.

Cheers,

Eain A Murphy
---

## [Editor Report · Acceptance letter]

30 Mar 2020

Dear Dr. Stern-Ginossar,

We are delighted to inform you that your manuscript, "Human cytomegalovirus long noncoding RNA4.9 regulates viral DNA replication," has been formally accepted for publication in PLOS Pathogens.

Best regards,

Kasturi Haldar

Editor-in-Chief

PLOS Pathogens

orcid.org/0000-0001-5065-158X

Michael Malim

Editor-in-Chief

PLOS Pathogens

orcid.org/0000-0002-7699-2064